# Boomerang Distillation Enables Zero-Shot Model Size Interpolation

**Sara Kangaslahti[1], Nihal V. Nayak[1]\*, Jonathan Geuter[1,2]\*, Marco Fumero[3],**
**Francesco Locatello[3], David Alvarez-Melis[1,2]**
[1]Harvard University    [2]Kempner Institute    [3]IST Austria
sarakangaslahti@g.harvard.edu

## Abstract

Large language models (LLMs) are typically deployed under diverse memory and compute constraints. Existing approaches build model families by training each size independently, which is prohibitively expensive and provides only coarse-grained size options. In this work, we identify a novel phenomenon that we call *boomerang distillation*: starting from a large base model (the *teacher)*, one first distills down to a small student and then progressively reconstructs intermediate-sized models by re-incorporating blocks of teacher layers into the student—*without any additional training*. This process produces zero-shot interpolated models of many intermediate sizes whose performance scales smoothly between the student and teacher, often matching or surpassing pretrained or distilled models of the same size. We further analyze when this type of interpolation succeeds, showing that alignment between teacher and student through pruning and distillation is essential. Boomerang distillation thus provides a simple and efficient way to generate fine-grained model families, dramatically reducing training cost while enabling flexible adaptation across deployment environments. The code and models are available at https://github.com/dcml-lab/boomerang-distillation.

## 1 Introduction

As large language models (LLMs) become integral to various applications, the challenge of adapting them efficiently to diverse hardware and deployment constraints is increasingly pressing. These models are now used in a wide variety of settings, ranging from edge devices (Narayan et al., 2025) to large-scale clusters (Comanici et al., 2025). Real-world deployment requires balancing multiple constraints, such as compute resources, energy consumption, and the trade-off between accuracy and latency (Huyen, 2022; Wu et al., 2022; Khandelwal et al., 2025). To address these diverse requirements, model developers increasingly release families of LLMs spanning different parameter scales (Team et al., 2024; Grattafiori et al., 2024; Yang et al., 2025). However, producing such model families remains highly resource-intensive. Conventional pretraining pipelines require enormous compute, making it impractical to train many variants from scratch. As a result, existing families typically include only a small set of coarse-grained model sizes, leaving significant gaps in the trade-off space between efficiency and capability. In this work, we investigate cost-efficient methods to construct pretrained LLM families with fine-grained size increments, enabling smoother adaptation to heterogeneous deployment constraints.

Knowledge distillation has become a common approach for producing LLM families of different sizes (Muralidharan et al., 2024). Rather than pretraining each model from scratch, practitioners often distill a pretrained *teacher* model into smaller *student* models (Hinton et al., 2015). Student models may be initialized either randomly or using parameter reduction techniques such as layer dropping (Men et al., 2024; Chen et al., 2025) or neuron pruning (Ma et al., 2023). They are then trained on large text corpora with distillation objectives, often combined with additional alignment losses such as cosine similarity or L2 distance. This paradigm is significantly more compute-efficient than independent training, reducing both FLOPs and the number of training tokens

---

\*Equal contribution.

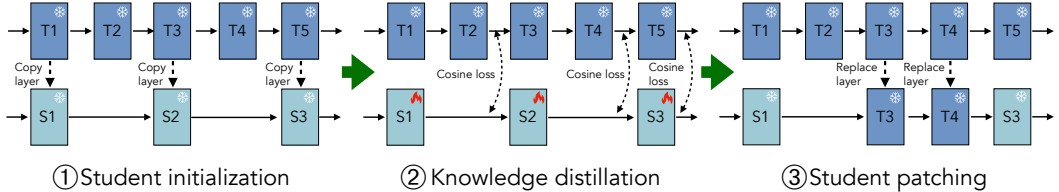

Figure 1: Overview of boomerang distillation. ① In this example, the student model is initialized by dropping layers from the pretrained teacher model. ② The teacher model is distilled into the student model with cross-entropy loss, knowledge distillation loss, and cosine distance loss (Equation 1). ③ After training the student model, a block of teacher layers corresponding to a student layer is inserted back into the model to get the zero-shot interpolated model.

required (Muralidharan et al., 2024). However, its key limitation is that each student still requires a full training run. As a result, scaling to fine-grained model sizes remains prohibitively expensive.

In this work, we identify a surprising phenomenon we call *boomerang distillation* (Figure 1): starting from a large teacher model, one can first distill down to a small student and then progressively reconstruct larger models by re-incorporating subsets of teacher layers into the student. This procedure yields a spectrum of intermediate model sizes **without any additional training**. Remarkably, these hybrids consistently achieve performance that interpolates smoothly between the student and teacher across downstream tasks (Figure 2). Unlike existing pruning-based approaches, which only use information from the teacher, boomerang distillation leverages both student and teacher information to form *true* interpolations between them. As a result, it consistently yields models that substantially outperform naive layer dropping and more advanced pruning techniques. In short, boomerang distillation reveals that zero-shot model size interpolation is not only possible, but also highly effective.

We conduct extensive experiments and ablations to characterize this phenomenon. First, we show that boomerang distillation only emerges when the student model is initialized from teacher weights and trained with a distillation objective plus an alignment loss such as cosine distance (§3.1). The resulting interpolated models match or exceed the performance of tailored distilled models of the same—or even larger—size (§3.2), with the alignment loss playing a critical role in the stability of boomerang distillation (§3.3). We further show that boomerang distillation generalizes to existing distilled models such as DistilBERT (Sanh et al., 2019) when combined with BERT (Devlin et al., 2019) (§3.4). Finally, we demonstrate that boomerang distillation-based models consistently outperform pruning methods across a variety of settings (Men et al., 2024; Yang et al., 2024) (§3.5) and provide extensive ablations aimed at understanding the impact of training data budgets and layer selection strategies (§3.6).

Our work makes the following contributions:

- We introduce *boomerang distillation*, a general phenomenon in model distillation that enables the creation of a family of models spanning student and teacher sizes *without any additional training* by patching the student with blocks of teacher layers (§2). These models smoothly interpolate size and performance between the student and teacher (§3.1). To our knowledge, this is the first study to identify and analyze this phenomenon and its zero-shot interpolation capabilities.

- We show that these interpolated models achieve performance on par with, and in some cases surpass, standard distilled models of the same size (§3.2). We also demonstrate the phenomenon across open-source models such as DistilBERT and BERT, highlighting its generality (§3.4).

- We perform thorough experiments to understand the conditions under which boomerang distillation arises (§3.3, Appendices E, F, G, H, I, and J) and demonstrate its consistent advantages over existing pruning-based approaches (§3.5). For example, we show that alignment loss, such as cosine distance loss, enables us to create boomerang distilled models with stable performance.

## 2 BOOMERANG DISTILLATION: KNOWLEDGE DISTILLATION WITH STUDENT PATCHING

We now describe the procedure underlying boomerang distillation. It consists of three key stages: (1) student initialization, (2) knowledge distillation, and (3) student patching (Figure 1).

**Preliminaries.** We consider the problem of distilling a pretrained transformer-based language model (teacher) into a smaller student model. Let the teacher LLM $\boldsymbol{T}$ have $N$ transformer layers, and the student model from the same family $\boldsymbol{S}$ have $M < N$ layers. We denote the parameters of the teacher and student models, respectively, as $\boldsymbol{\theta}_T = (\boldsymbol{\theta}_T^E, \boldsymbol{\theta}_T^{(1)}, \ldots, \boldsymbol{\theta}_T^{(N)}, \boldsymbol{\theta}_T^D)$ and $\boldsymbol{\theta}_S = (\boldsymbol{\theta}_S^E, \boldsymbol{\theta}_S^{(1)}, \ldots, \boldsymbol{\theta}_S^{(M)}, \boldsymbol{\theta}_S^D)$, where $\boldsymbol{\theta}^{(i)}$ represents the $i$-th transformer block, and $\boldsymbol{\theta}^E$ and $\boldsymbol{\theta}^D$ denote the embedding layer and LM head, respectively. All student and teacher layers produce hidden states of the same dimension. We assume access to corpus $\mathcal{X}$ to train the student model using a knowledge distillation objective; but do not assume access to the teacher's pretraining data, consistent with realistic settings (Yang et al., 2025). Our goal is to learn $\boldsymbol{\theta}_S$ such that after training, for any nonnegative $K$ with $M + K < N$, we can deterministically construct an intermediate model $\boldsymbol{\theta}_I$ with $M + K$ layers from $(\boldsymbol{\theta}_S, \boldsymbol{\theta}_T)$.

## 2.1 STUDENT INITIALIZATION

To initialize the student model, we partition the teacher's $N$ transformer layers into $M$ contiguous blocks $\mathcal{B} = (\mathbf{b}^{(1)} \ldots \mathbf{b}^{(M)})$, where the $i$-th block $\mathbf{b}^{(i)}$ consists of the layers $(\boldsymbol{\theta}_T^{(\ell_i)}, \ldots, \boldsymbol{\theta}_T^{(\ell_{i+1}-1)})$ for some indices $1 = \ell_1 < \cdots < \ell_M \leq N$ (with $\mathbf{b}^{(M)} \triangleq (\boldsymbol{\theta}_T^{(\ell_M)}, \ldots, \boldsymbol{\theta}_T^{(N)})$). Following prior work (Chen et al., 2025), we initialize the student as $\boldsymbol{\theta}_S^{(i)} = \boldsymbol{\theta}_T^{(\ell_i)}$, $i = 1, ..., M$, $\boldsymbol{\theta}_S^E = \boldsymbol{\theta}_T^E$, and $\boldsymbol{\theta}_S^D = \boldsymbol{\theta}_T^D$.

## 2.2 KNOWLEDGE DISTILLATION

The initialized student model is then trained via distillation to recover performance while remaining aligned to the teacher model, which will enable subsequent interpolation by patching the student model (Section 2.3).

Given a training sequence $x = (x_1, \ldots, x_L) \sim \mathcal{X}$ of $L$ tokens, let $\boldsymbol{z}_j^T = \boldsymbol{T}(x_{<j})$ and $\boldsymbol{z}_j^S = \boldsymbol{S}(x_{<j})$ be the logits of the teacher and student model for the $j$-th token. Following standard knowledge distillation approaches (Hinton et al., 2015; Muralidharan et al., 2024), in addition to the cross entropy loss $\mathcal{L}_{\mathrm{CE}}(x_j \mid x_{<j}; \boldsymbol{\theta}_S)$, we add a KL divergence loss:

$$\mathcal{L}_{\mathrm{KL}}(x_{<j}; \boldsymbol{\theta}_S) = \tau^2 \cdot \mathrm{KL}\big(\mathrm{softmax}\big(\boldsymbol{z}_j^T/\tau\big) \,\|\, \mathrm{softmax}\big(\boldsymbol{z}_j^S/\tau\big)\big)$$

where $\tau$ is a temperature parameter. To further align representations, we introduce a cosine distance loss (Sanh et al., 2019), which encourages the hidden states of the student across all layers to remain close to those of the teacher model. We refer to this as the *alignment loss*. The key idea is to ensure the student layer approximates the teacher block's output, so we can swap the teacher block back in to create interpolated models (§2.3). We align the hidden states of the $i$-th layer in the student model with the hidden states produced by teacher block $\mathbf{b}^{(i)}$, which corresponds to the $(\ell_{i+1} - 1)$-th layer in the teacher, using a cosine distance loss:

$$\mathcal{L}_{\cos}^{(i)}(x_{<j}; \boldsymbol{\theta}_S) = 1 - \left(\boldsymbol{x}_j^{(S,i)} \cdot \boldsymbol{x}_j^{(T,l_{i+1}-1)}\right) \Big/ \left(||\boldsymbol{x}_j^{(S,i)}||\, ||\boldsymbol{x}_j^{(T,l_{i+1}-1)}||\right)$$

where $\boldsymbol{x}_j^{(S,i)}$ and $\boldsymbol{x}_j^{(T,\ell_{i+1}-1)}$ are the hidden states of $i$-th layer of the student and $(\ell_{i+1}-1)$-th layer of the teacher for the $j$-th token given input $x_{<j}$.

The full training objective for the student is therefore:

$$\mathcal{L}(x, \boldsymbol{\theta}_S) = \mathcal{L}_{\mathrm{CE}}(x_j \mid x_{<j}; \boldsymbol{\theta}_S) + \lambda_{\mathrm{KL}}\, \mathcal{L}_{\mathrm{KL}}(x_{<j}; \boldsymbol{\theta}_S) + \lambda_{\cos} \sum_{i=1}^{M} \mathcal{L}_{\cos}^{(i)}(x_{<j}; \boldsymbol{\theta}_S) \tag{1}$$

where $\lambda_{\mathrm{KL}} > 0$ and $\lambda_{\cos} > 0$ are hyperparameters tuned to weigh the three loss terms (Appendix C).

## 2.3 STUDENT PATCHING

After distillation, we construct interpolated models by selectively *patching* the student with layers from the teacher model (Figure 1, step ③). Specifically, replacing the $i$-th student layer with its

corresponding block of teacher layers $\mathbf{b}^{(i)} = (\boldsymbol{\theta}_T^{(\ell_i)}, \ldots, \boldsymbol{\theta}_T^{(\ell_{i+1}-1)})$ yields:

$$(\boldsymbol{\theta}_S^{(1)}, \boldsymbol{\theta}_S^{(2)}, \cdots, \boldsymbol{\theta}_S^{(i-1)}, \boldsymbol{\theta}_S^{(i)}, \boldsymbol{\theta}_S^{(i+1)}, \cdots, \boldsymbol{\theta}_S^{(M)}) \rightarrow (\boldsymbol{\theta}_S^{(1)}, \boldsymbol{\theta}_S^{(2)}, \cdots, \boldsymbol{\theta}_S^{(i-1)}, \mathbf{b}^{(i)}, \boldsymbol{\theta}_S^{(i+1)}, \cdots, \boldsymbol{\theta}_S^{(M)})$$
$$= (\boldsymbol{\theta}_S^{(1)}, \boldsymbol{\theta}_S^{(2)}, \cdots, \boldsymbol{\theta}_S^{(i-1)}, \boldsymbol{\theta}_T^{(\ell_i)}, \boldsymbol{\theta}_T^{(\ell_i+1)}, \cdots, \boldsymbol{\theta}_T^{(\ell_{i+1}-1)}, \boldsymbol{\theta}_S^{(i+1)}, \cdots, \boldsymbol{\theta}_S^{(M)})$$

Applying this substitution repeatedly produces models of various intermediate sizes between $\boldsymbol{S}$ and $\boldsymbol{T}$. Once we have the set transformer layers for the interpolated model, we pick the embedding layer from the model that contributes the first layer (i.e., pick $\boldsymbol{\theta}_S^E$ when using $\boldsymbol{\theta}_S^{(1)}$, and $\boldsymbol{\theta}_T^E$ when using $\mathbf{b}^{(1)}$), and likewise pick the LM head from that model that contributes the last layer.

## 3 EXPERIMENTS

In this section, we study the boomerang distillation phenomenon in depth. We begin by identifying necessary conditions for it to succeed (§3.1). Then, we compare the quality of the zero-shot interpolated models created from boomerang distillation to models trained with standard knowledge distillation (§3.2). Next, we analyze the role of individual loss terms in enabling interpolation between student and teacher (§3.3). We further demonstrate that boomerang distillation also arises in existing pretrained models (§3.4). Then, we compare boomerang distillation to layer pruning methods, showing that our interpolated models perform significantly better than layer pruning approaches for the same model size (§3.5). Finally, we summarize additional experiments with boomerang distillation on the impact of training tokens and initial student model sizes (§3.6). In these experiments, we use Qwen3-4B-Base as our main teacher model. In Appendices F, G, and H, we reproduce experiments from Sections 3.1, 3.2, and 3.3 with Pythia-2.8B and Llama-3.2-3B as the teacher models and report similar findings, demonstrating that boomerang distillation is a general phenomenon in LLMs.

**Boomerang Distillation Implementation Details.** We primarily use Qwen3-4B-Base (Yang et al., 2025) as the teacher model. The student model, with 2.7B inference-time parameters, is initialized by removing every other layer (except the last layer) from the teacher model and is then trained on the deduplicated Pile (Gao et al., 2021) using the overall loss (Equation 1) with a budget of 2.1B tokens. To create interpolated models, we patch the distilled student model with corresponding contiguous blocks of teacher layers in reverse order, starting from the last layer. In all experiments, we report the inference-time parameters as the parameter count. For more details on student initialization, training, and patching order for all pretrained teacher models, see Appendix B.

**Datasets.** We use the same classification and generation datasets throughout the paper. We use `lm-evaluation-harness` (Gao et al., 2023) to evaluate all of the models and report classification accuracy on ten datasets and exact match accuracy on three generation datasets. We also compute perplexity on the WikiText dataset (Merity et al., 2017) for all models and report it in Appendix M.1. For more details on datasets, see Appendix D.

### 3.1 THE BOOMERANG DISTILLATION PHENOMENON

In this section, we study conditions necessary for boomerang distillation to occur, and demonstrate its strong interpolation performance between the student and teacher model.

**Setup.** We evaluate against two key baselines: (1) naive layer pruning and (2) distillation with a randomly initialized student model. In naive layer pruning, we iteratively remove layers from the teacher model, starting with the second layer and then every other layer (up to $\boldsymbol{\theta}_T^{(N-2)}$) until the desired model size is attained. This corresponds to the same set of teacher layers used in the distilled and patched student model, but without any distillation training. This baseline tests if knowledge distillation (2.2) is essential for teacher patching. For the second baseline, distillation with a randomly initialized student model, instead of initializing the student from teacher layers, we initialize all weights randomly (leaving the architecture unchanged) before distilling with the same loss from Equation 1. This baseline tests if student initialization with teacher weights (2.1) is required for student patching to create models with interpolated performance.

**Results.** Figure 2 shows that boomerang distillation creates models whose size and performance interpolate smoothly (for a complete breakdown, see Appendix Figures 31 and 35). This enables

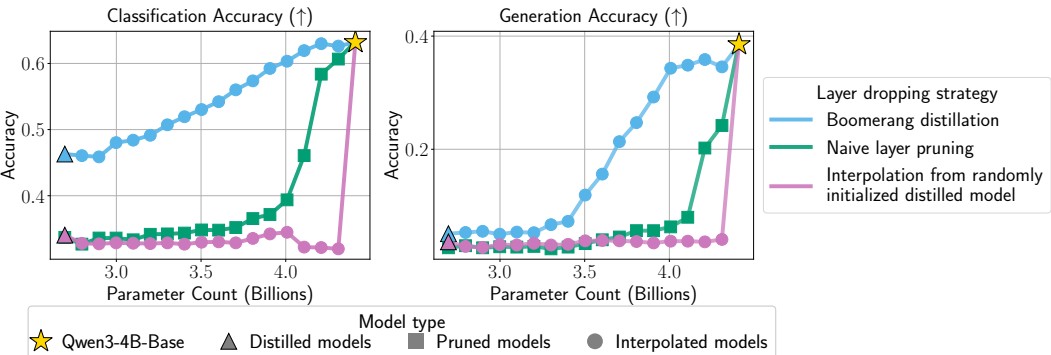

Figure 2: **Boomerang distillation produces models with smooth size–performance interpolation**, consistently outperforming naive layer pruning and interpolation from randomly initialized distilled models. These results indicate that effective interpolation depends on initializing the student with teacher weights and training under a knowledge distillation objective.

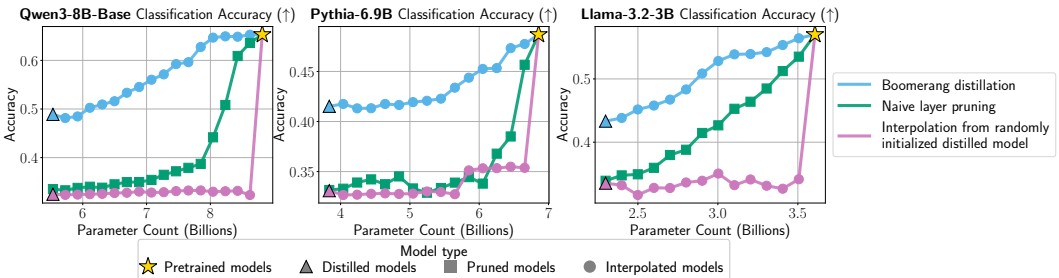

Figure 3: **Boomerang distillation emerges across model families**. Shown here for Qwen3-8B, Pythia-6.9B, and Llama-3.2-3B, boomerang distillation yields intermediate models with smooth accuracy–parameter scaling, outperforming naive layer pruning and random interpolation baselines.

us to create a full suite of intermediate models without any additional training. We show that boomerang distillation occurs when the layer-pruned, distilled student model is patched with corresponding teacher layers. In contrast, we find that the boomerang distillation phenomenon does not occur for naive layer pruning and randomly initialized distillation baselines. When we naively drop layers, there is a significant drop in classification and generation performance for models of size less than 4B inference-time parameters. However, we do not see such a dramatic drop in performance in the interpolated models created with boomerang distillation. In the randomly initialized model, there is almost no gain in performance when patching teacher layers to the distilled student. These results show that layer pruning or distillation alone is not sufficient for boomerang distillation. We also observe that boomerang distillation shows smoother interpolation in classification accuracy than in generation accuracy. This discrepancy has been observed in prior work on layer pruning: Short-GPT (Men et al., 2024) hypothesizes that errors accumulate much more in smaller pruned models than in larger ones. However, in Section 3.5, we show that boomerang distillation maintains much higher generation performance for smaller models compared to pruning methods.

In Figure 3, we show that boomerang distillation also occurs in Qwen3-8B-Base, Pythia-6.9B, and Llama-3.2-3B, demonstrating that boomerang distillation is a general phenomenon in distilled LLMs that can be observed across various model sizes and families (See Appendix F for full results). We note that in the Llama model, we keep the first two layers instead of the last two layers during student initialization and patch the model starting from the first layer. This is because the first two layers of the teacher model have low cosine similarity with each other, and excluding them from the training hurts the performance of the student model and the interpolated models (see Appendix I for cosine similarity analysis).

## 3.2 How Good is Boomerang Distillation?

To test the quality of the zero-shot interpolated models created using boomerang distillation, we compare them against models of intermediate sizes created through standard knowledge distillation.

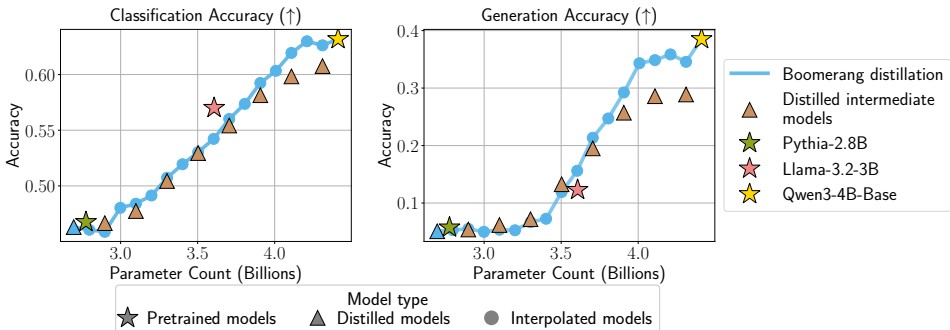

Figure 4: **Interpolated models produced using boomerang distillation have comparable performance to pretrained and standard distilled models.** We compare the interpolation performance of boomerang distillation to distilled models initialized with the corresponding teacher layers and distilled using Equation 1. At small sizes, the interpolated models have comparable performance to distilled and pretrained models. At larger sizes, the interpolated models outperform distilled models, likely due to catastrophic forgetting caused by distilling on a presumably lower-quality corpus.

**Setup.** For standard knowledge distillation, we follow the training setup in Appendix B to train intermediate-size models. We initialize the intermediate models by removing every other teacher model layer starting from the second layer and continuing up to layer $\theta_T^{(j)}$ to match the set of layers in the same size interpolated model. For a fair comparison, we train the intermediate model with our overall loss (Equation 1) for 2.1B tokens. To contextualize these results, we also compare with pretrained LLMs: Pythia-2.8B (Biderman et al., 2023) and Llama-3.2-3B (Grattafiori et al., 2024).

**Results.** Boomerang distillation produces interpolated models that show comparable performance to the intermediate models trained via standard knowledge distillation, even outperforming them at larger sizes (Figure 4; for a per-task breakdown, see Appendix Figures 32 and 36). A key difference between the models from boomerang distillation and the standard distilled models is that we only need to train a single small student model and create interpolated models by patching teacher weights without additional training. This dramatically reduces the time and resources needed to create a family of intermediate-sized models by orders of magnitude.

We also observe that the interpolated models achieve comparable performance to existing pretrained models. Despite the student model being trained on far fewer tokens than Pythia-2.8B and Llama-3.2-3B, boomerang distillation adaptively produces interpolated models of comparable size and performance without any additional training. Finally, in Appendices G and H, we also find that interpolated models created with boomerang distillation using Pythia and Llama achieve comparable performance to distilled models. This demonstrates the universality of the boomerang distillation phenomenon across different model families.

In Figure 4, we observe that the intermediate models at larger sizes underperform boomerang distillation models. We suspect that updating the weights of Qwen3-4B-Base on a presumably lower-quality corpus, such as The Pile, leads to catastrophic forgetting (French, 1999; Kirkpatrick et al., 2017), which results in a drop in performance. This is a practical problem with open-weight models because we often do not have access to the original training corpus (Jiang et al., 2023; Yang et al., 2025). We also show that such a drop in performance also occurs for intermediate distilled models for Llama-3.2-3B (Figure 20 in Appendix H). On the other hand, we find that intermediate models created by distilling Pythia-2.8B (Figure 17 in Appendix G) perform better than interpolated models since the Pythia models are trained on The Pile. These results suggest that boomerang distillation can retain the benefits of the original model, trained on a higher-quality corpus, by patching its weights back into the student model.

### 3.3 EFFECT OF KNOWLEDGE DISTILLATION

In this experiment, we aim to understand which of the losses in the knowledge distillation objective contribute to the boomerang distillation phenomenon.

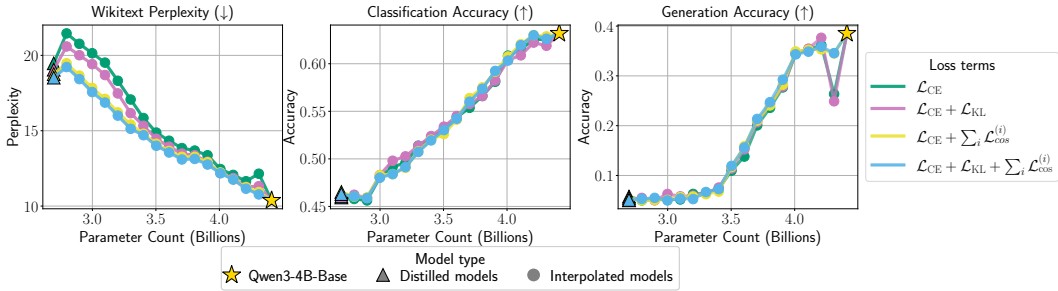

Figure 5: **Per-layer loss yields stable and smoother interpolation performance.** Models distilled with per-layer cosine distance loss have smoother interpolation behavior across all model sizes. However, boomerang distillation still occurs for models without per-layer cosine distance loss, indicating that initialization using teacher layers provides substantial alignment information.

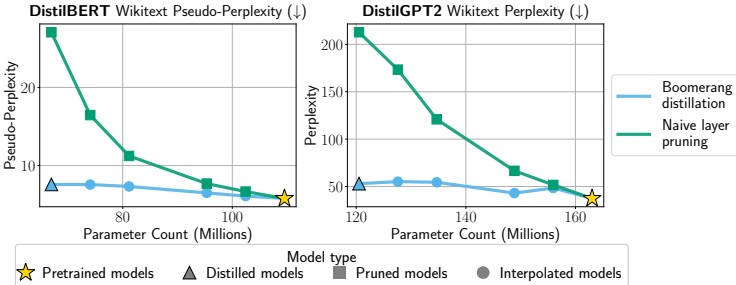

Figure 6: **Boomerang distillation works for off-the-shelf pretrained models without any additional training.** Boomerang distillation via student patching DistilBERT (Sanh et al., 2019) with BERT layers (Devlin et al., 2019) (left) and student patching DistilGPT2 (Sanh et al., 2019) with GPT2 layers (Radford et al., 2019) (right) produces interpolated models that significantly outperform naive layer pruning from the teacher model.

**Setup.** We compare four loss terms in this experiment: (1) cross entropy ($\mathcal{L}_{\text{CE}}$), (2) cross entropy with knowledge distillation loss ($\mathcal{L}_{\text{CE}} + \mathcal{L}_{\text{KL}}$), (3) cross entropy with alignment loss ($\mathcal{L}_{\text{CE}} + \sum_i \mathcal{L}_{cos}^{(i)}$), (4) overall loss, i.e., cross entropy with knowledge distillation loss and alignment loss ($\mathcal{L}_{\text{CE}} + \mathcal{L}_{\text{KL}} + \sum_i \mathcal{L}_{cos}^{(i)}$). We follow the setup from Appendix B to initialize the student models and train on 2.1B tokens with different loss objectives.

**Result.** Figure 5 shows that the cross entropy with knowledge distillation loss and alignment loss (Equation 1) creates interpolated models with the lowest perplexity compared to the other loss terms (for full per-task breakdown see Appendix Figures 33 and 37). We do not see a meaningful difference in classification and generation accuracies on downstream tasks for the majority of model sizes. However, at both extremes of intermediate model size (leftmost and rightmost interpolated models), there is slight instability in performance, especially for the cases with no per-layer loss. These models correspond to patching the last few and first few teacher layers, respectively, indicating that layer-wise alignment is especially important for the first and last layers. This aligns with prior work showing that the initial and last model layers are distinct, while intermediate layers are more interchangeable (Gromov et al., 2024; Men et al., 2024). In Appendices G and H, we demonstrate that Pythia and Llama models produce a similar ranking of loss objectives by perplexity, while also showing meaningful differences in classification accuracy.

While these results confirm that alignment losses, such as cosine distance loss, are needed to achieve the best-performing interpolated models, we still observe boomerang distillation even when students are trained with only a cross entropy objective. This suggests that initializing the student with teacher weights is itself a central factor in enabling boomerang distillation, consistent with our findings in Section 3.1. An open question, however, is whether comparable performance and stability can be achieved *without* retaining the teacher weights in memory, which would substantially reduce the memory footprint.

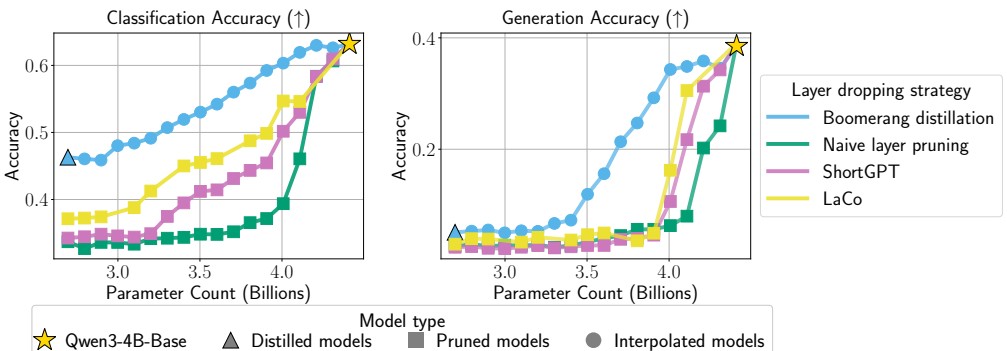

Figure 7: **Boomerang distillation performs significantly better than layer pruning methods.** We compare boomerang distillation to two popular layer pruning approaches, LaCo (Yang et al., 2024) and ShortGPT (Men et al., 2024). Boomerang distillation has significantly better performance across all intermediate sizes, especially for generation tasks, where layer pruning quickly degrades to very low accuracy.

## 3.4 ZERO-SHOT MODEL SIZE INTERPOLATION WITH EXISTING OFF-THE-SHELF MODELS

Here we show that boomerang distillation occurs even between popular existing off-the-shelf open-source models and their distilled variants (Devlin et al., 2019; Radford et al., 2019; Sanh et al., 2019).

**Setup.** We interpolate between off-the-shelf DistilBERT and BERT, and DistilGPT2 and GPT2. Similar to our setup, DistilBERT and DistilGPT2 are initialized by pruning alternate layers from their teacher models, BERT and GPT2, and then trained with knowledge distillation and cosine distance loss objective. Although DistilBERT and DistilGPT2 use cosine distance loss only on the final hidden states, we use both models without modification. We then add back the teacher layers to patch the distilled student models to create the interpolated models. We report the perplexity for both models, as they do not exhibit strong out-of-the-box zero-shot performance. We report pseudo-perplexity for BERT and DistilBERT and perplexity for GPT2 and DistilGPT2 on WikiText.

**Results.** Figure 6 shows that intermediate models created by patching corresponding teacher layers from BERT into DistilBERT show a clean interpolation in performance without any training. We observe that the intermediate GPT2 models show a less clean interpolation compared to the BERT models, yet they still outperform the naive layer pruning baseline. This result shows that the boomerang distillation phenomenon occurs even in existing small pretrained language models. To our knowledge, we are the first to discover zero-shot model size interpolation between these models.

Many existing LLMs distilled from larger teacher models are not readily usable for boomerang distillation as their setup differs from ours in several ways. For example, Muralidharan et al. (2024) uses layer pruning along with neuron pruning to initialize the student model, which creates a mismatch in the dimensions of the hidden state. This prevents us from patching the student model with teacher weights. Furthermore, existing distillation frameworks often do not use cosine distance loss in their training (Kim et al., 2024; Gu et al., 2024). We suspect this is because it increases the memory footprint during training and does not significantly improve the student model performance. Distilling large-scale LLMs with layer pruning and cosine distance loss, with a large token budget, for boomerang distillation is a promising direction for future research.

## 3.5 COMPARISON TO LAYER PRUNING METHODS

We compare boomerang distillation against layer pruning approaches, since they most closely approximate our setting of creating models with different numbers of layers without additional training.

**Setup.** We consider two popular layer pruning methods: Layer Collapse (LaCo) (Yang et al., 2024) and ShortGPT (Men et al., 2024). LaCo identifies blocks of layers with high cosine similarity between the outputs of the first and last layer in the block, then collapses the later layers in the block into the first one by adding their difference in parameters. ShortGPT ranks each layer in the model by its block influence, or cosine distance between the input and output activations of the layer. It then prunes the layers with the lowest block influence score. For implementation details, see Appendix N.

**Results.** Figure 7 shows that zero-shot interpolation via boomerang distillation results in significantly better classification and generation accuracy than layer pruning methods (for full breakdown see Appendix Figures 34 and 38). In particular, as observed in the ShortGPT paper (Men et al., 2024), the generation capabilities for the pruning approaches collapse to near zero after just a few layers removed, whereas boomerang distillation maintains higher generation accuracy for much smaller models. Boomerang distillation also smoothly interpolates in classification accuracy between the distilled student model and the teacher model, whereas the classification accuracy of all three pruning methods plateaus to near random performance for models of size around 3B parameters. We note that both of these layer pruning strategies could be used to initialize the student model in a boomerang distillation pipeline and leave exploration of different layer pruning initializations for boomerang distillation to future work.

## 3.6 ABLATIONS

We include additional experiments in Appendix E. An ablation on smaller student models, achieved by a more aggressive layer pruning, shows that boomerang distillation performs well as long as the distilled student model has non-trivial performance on target tasks. Furthermore, we study the effect of training tokens on boomerang distillation and find that increasing the student's training budget yields interpolated models with improved performance.

## 4 RELATED WORK

**Model Interpolation.** Model interpolation is a key technique that combines trained models by directly interpolating their weights (Singh & Jaggi, 2020; Frankle et al., 2020; Wortsman et al., 2022). These works focus on combining weights of multiple models with additional training to improve robustness and out-of-domain generalization (Wortsman et al., 2022; Jin et al., 2023; Dang et al., 2025), create multi-task models (Ilharco et al., 2023; Yadav et al., 2023; Zhu et al., 2025), and controllable generation (Gandikota et al., 2024; Kangaslahti & Alvarez-Melis, 2024). All these works interpolate between model weights of the same size. In contrast, we interpolate between the student and the teacher model to create interpolated models of different sizes. Cai et al. (2025) trains the teacher model in an elastic transformer architecture along with a Gumbel Softmax-based router and interpolates model sizes using the trained router. On the other hand, boomerang distillation trains a smaller student model using a standard knowledge distillation pipeline and interpolates model sizes by patching the distilled student with teacher layers, and does not require training a specialized router.

**Knowledge Distillation.** Knowledge distillation is a popular method used to train a smaller student model to mimic the behavior of the larger teacher model with fewer parameters (Hinton et al., 2015; Sanh et al., 2019). Knowledge distillation can be used to train a smaller student model even if the teacher and the student do not share the same architecture. This has enabled researchers to distill vision models (Oquab et al., 2023), large language models (Team et al., 2024), and proprietary API-based models (Taori et al., 2023; Gudibande et al., 2024) into smaller models. Recently, knowledge distillation has been used to distill a pretrained teacher LLM into multiple smaller student LLMs of varying sizes to create a family of language models, but this approach incurs significant compute cost (Muralidharan et al., 2024; Sreenivas et al., 2024). Alternatively, existing work in the vision domain (Bai et al., 2020; Shen et al., 2020) has proposed to reduce the cost of training student models by aligning student and teacher layers in a few-shot setting, but they are limited to small-scale models and datasets and require retraining for each model size. In contrast to these approaches, we use knowledge distillation to train a single student LLM using a larger teacher LLM and create interpolated models of fine-grained sizes without requiring any additional training.

**Pruning.** Model pruning is a widely studied area where the goal is to compress model parameters by removing redundant parameters to reduce computational requirements while maintaining the performance of the full model (LeCun et al., 1989; Han et al., 2015; Sun et al., 2024). Several techniques have been proposed to prune model parameters. These include layer dropping (Men et al., 2024; Chen et al., 2025), neuron pruning (Ma et al., 2023), SVD-based pruning (Yuan et al., 2023; Lin et al., 2024; Wang et al., 2025), and more (Cheng et al., 2024). They often require training the pruned model over an auxiliary dataset to recover the initial performance (Xia et al., 2024). In this work, we initialize a student model by dropping layers from an existing pretrained large language model and then train it with a knowledge distillation objective.

**Dynamic Compute Allocation.** Dynamically allocating a variable amount of compute at inference time based on task complexity is critical for today's intelligent systems (Damani et al., 2024). Several techniques, such as early exiting (Schuster et al., 2022; Elhoushi et al., 2024), test-time scaling (Snell et al., 2024; Muennighoff et al., 2025), and compute-adaptive embeddings (Kusupati et al., 2022; Lee et al., 2024), have been proposed for dynamic compute allocation. In our work, we focus on dynamically creating new models by interpolating model sizes that require different amounts of compute during inference. Existing approaches that produce models of variable sizes often require explicit training (Kusupati et al., 2022; Lee et al., 2024), which is expensive and time-consuming when many fine-grained model variants are needed. In contrast, we create fine-grained interpolated models without any additional training with only one student model.

## 5 Conclusion

We identify boomerang distillation, a novel phenomenon in large language models. We show that boomerang distillation can be used to create a family of models that smoothly interpolate in size and performance between a given student and teacher model, *without any additional training*. In our experiments, we show that boomerang distillation occurs when training a student model with knowledge distillation from a teacher model. Crucially, we identify that the student has to be initialized from the teacher with layer pruning. Furthermore, we observe that boomerang distillation occurs even in existing open-source models such as DistilBERT and DistilGPT2 (Sanh et al., 2019). Our interpolated models consistently match or even outperform models of the same size directly trained with knowledge distillation, and exhibit superior downstream performance compared to existing pruning approaches. In conclusion, we provide a simple recipe for creating fine-grained model families from a single student-teacher pair, which significantly reduces training time and cost.

## Acknowledgments

We thank Yonatan Belinkov, Bingbin Liu, Lyndon Lam, Chloe Su, and the members of the ML Foundations group and the Kempner Institute for their thoughtful feedback on the manuscript. David Alvarez-Melis, Sara Kangaslahti, Jonathan Geuter, and Nihal V. Nayak acknowledge support from the National Science Foundation Graduate Research Fellowship (Grant No. DGE 2140743), the Kempner Institute, FAS Dean's Competitive Fund for Promising Scholarship, Aramont Fellowship Fund, and the NSF AI-SDM Institute (Grant No. IIS-2229881). Francesco Locatello's contribution to this research was funded in part by the Austrian Science Fund (FWF) 10.55776/COE12.

## Reproducibility statement

We implement all our experiments using PyTorch (Paszke et al., 2019) and HuggingFace transformers (Wolf et al., 2019) packages. We also experiment with public models available on HuggingFace Hub. We provide our code and models at `https://github.com/dcml-lab/boomerang-distillation/`.

## Ethics statement

Interpolated models created using boomerang distillation may inherit or amplify the biases of the pretrained teacher model. Before deploying, we recommend comprehensively evaluating the models on the target tasks to identify potential biases. To further mitigate any residual biases, we suggest training the model to follow instructions and carry out additional safety training.

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

CONTENTS

## A    LIMITATIONS AND FUTURE WORK

We briefly discuss the limitations of boomerang distillation and directions for future work.

Boomerang distillation requires a distilled student LLM, which can be computationally expensive to train. As discussed in Section 3.1, we show that a distilled student LLM trained is crucial for boomerang distillation. While we get interpolated models of intermediate sizes without any additional training, training the student LLM itself requires a significant amount of compute.

Our computational resources limit the model size and number of distillation tokens in our experiments. Scaling this approach to larger models with a greater token budget is an exciting avenue for future work.

Boomerang distillation could also benefit from a more sophisticated initialization and student patching order. While we initialize the student model with every other layer in Section 3, we show in Appendix E.2 that initializing the student in a manner that maintains alignment allows for boomerang distillation with even smaller student models. In our work, we consider two approaches to patching the student model: either starting from the first layers or the last layers. However, in some cases, this naive patching order can lead to instability in performance in the interpolated models (Appendix I). In Appendix I, we analyze per-layer activation cosine similarity between all pairs of the distilled student and the base teacher layers in Llama-3.2-3B. We found that patching the student model with layers that have low cosine similarity to their teacher layers provides smoother interpolation performance. Therefore, initializing and patching the student models in a manner that is guided by the similarity of the layers could help mitigate the instability of the interpolated models.

Boomerang distillation requires the distilled student to be created via layer pruning. Combining this with other pruning strategies, such as width pruning and attention head pruning, may not work out of the box: If the teacher and student have different hidden dimensions due to width pruning, student patching cannot be applied out of the box because it would result in a mismatch in the output dimension of the hidden states. A similar obstacle occurs when pruning attention heads. Extending boomerang distillation in these settings is a promising future direction.

Finally, although we show that boomerang distillation is a phenomenon that occurs in language models, it remains to be seen whether it also occurs in other modalities, such as vision (Siméoni et al., 2025) and audio (Radford et al., 2023), that use transformer-based architectures. We leave extensions to other modalities as future work.

## B    BOOMERANG DISTILLATION IMPLEMENTATION

For all the experiments in Section 3, we primarily consider Qwen3-4B-Base (Yang et al., 2025) as a teacher model. We follow the same student initialization and training setup for additional models in Appendix F. All the implementation was done using PyTorch (Paszke et al., 2019) and HuggingFace transformers (Wolf et al., 2019).

| Teacher Model | | Student Model | |
|---|---|---|---|
| Name | Inf. params | Train. params | Inf. params |
| Qwen3-4B-Base | 4.4B | 2.3B | 2.7B |
| Qwen3-8B-Base | 8.8B | 4.9B | 5.6B |
| Pythia-2.8B | 2.8B | 1.6B | 1.6B |
| Pythia-6.9B | 6.9B | 3.8B | 3.8B |
| Llama-3.2-3B | 3.6B | 1.9B | 2.3B |

Table 1: **The sizes of the initialized student models after pruning the layers from the teacher model.** We note that the Pythia models do not employ weight tying, so their train and inference parameters are equivalent. On the other hand, the Qwen and Llama models weight tie their embedding layers and LM heads, so their inference-time parameters are higher than their training parameters. This is because both the embedding layer and LM head are used during inference.

**Student Initialization.** For convenience and increased granularity, in our experiments, similar to Sanh et al. (2019), we drop every other layer from the teacher model to initialize the student model. However, our work is not limited to this setting and could benefit from informed initialization strategies (Men et al., 2024). We also keep the last teacher layer, since doing so has been shown to be essential when pruning (Gromov et al., 2024). Table 1 summarizes the trainable and inference parameters of the teacher and the student models. In Qwen and Llama models, the number of trainable and inference-time parameters differs because the embedding layer is reused as the language modeling head during training. In all experiments, we report the inference-time parameters as the parameter count.

**Training.** We distill the student model on 2.1B tokens of the deduplicated Pile (Gao et al., 2021) using the overall loss (Eq. 1). We train the models on four NVIDIA H100 GPUs or four H200 GPUs, depending on their availability. Based on the size of the student model, the total training time typically ranged from 12 to 72 hours. Unless stated otherwise, we use the same hyperparameters to train all the student models. For full training hyperparameters, see Appendix C.

**Student Patching.** We perform student patching by replacing each student layer with its corresponding block of teacher layers. For all models except the Llama models, we patch the student layers starting backwards from the $M$-th layer and progressively patch more layers until all the layers are replaced with the teacher blocks. For the Llama models, we patch starting from the first layer and progressively patch until the $M$-th layer (see Appendix I for more details). As mentioned in Section 2.3, depending on the order of patching, we use the embedding and language modeling differently. In Qwen and Pythia models, we use the embedding layer from the distilled student model and the language modeling head from the teacher model. In Llama, we use the embedding layer from the teacher model and the language modeling head from the distilled teacher model.

## C Hyperparameters

| Hyperparameters | Values |
|---|---|
| Learning rate | 3e-4 |
| Learning rate scheduler | cosine |
| Warmup ratio | 0.01 |
| Optimizer | AdamW |
| Adam betas | (0.9, 0.95) |
| Adam epsilon | 1e-8 |
| Weight decay | 0.1 |
| Max. gradient norm | 1.0 |
| Number of training steps | 500 |
| Max. sequence length | 2048 |
| Effective batch size | 2048 |
| Mixed precision | bf16 |
| KLDiv weight $\lambda_{\mathrm{KL}}$ | 0.1 |
| Cosine distance weight $\lambda_{\mathrm{cos}}$ | 2.0 / (M+1) |

Table 2: **Hyperparameters used to train the student model.** We choose the training hyperparameters to align with the values used in Pythia training (Biderman et al., 2023) and set the KLDiv and cosine distance weights such that the cross entropy, KLDiv, and cosine distance loss are approximately equal in magnitude at the beginning of training.

Table 2 lists all the hyperparameters used to train the student model.

## D Datasets

We utilize the same evaluation datasets throughout the paper. We use `lm-evaluation-harness` (Gao et al., 2023) to evaluate classification accuracy, generation exact match accuracy, and perplexity. We compute classification accuracy on 10 tasks: ARC-easy

and ARC-challenge (Clark et al., 2018), BoolQ (Clark et al., 2019), HellaSwag (Zellers et al., 2019), OpenBookQA (Mihaylov et al., 2018), PIQA (Bisk et al., 2020), WinoGrande (Sakaguchi et al., 2020), RACE (Lai et al., 2017), MMLU (Hendrycks et al., 2021a), and RTE (Wang et al., 2019). For generation, we report exact match accuracy on 3 tasks: GSM8K (Cobbe et al., 2021), IFEval (Zhou et al., 2023), and MATH (Hendrycks et al., 2021b). We also compute perplexity on the WikiText dataset (Merity et al., 2017) for all experiments and report it in Appendix M.1.

## E    ADDITIONAL ABLATION EXPERIMENTS

### E.1    ABLATING DISTILLED MODEL SIZES

In this experiment, we test what size of distilled student model is best for boomerang distillation. Ideally, the student model should be as small as possible while maintaining interpolation performance.

**Setup.**    We train four student models initialized by keeping every other layer, every third layer, every fourth layer, and every fifth layer of the teacher model.

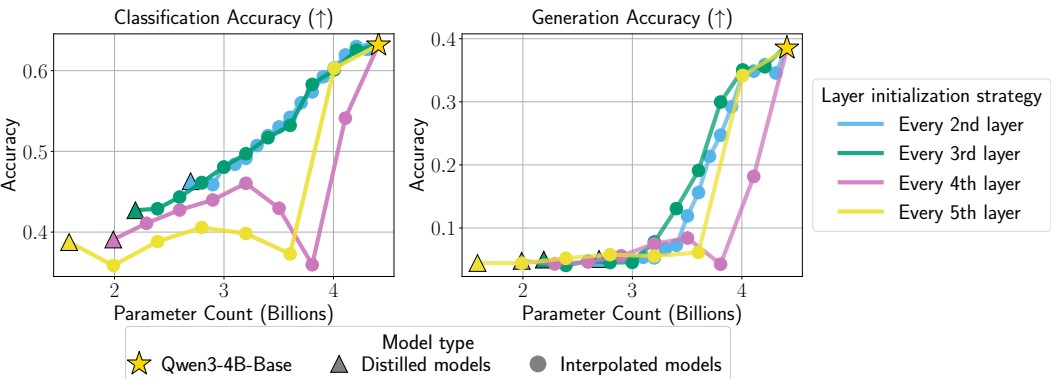

Figure 8: **Boomerang distillation occurs for smaller student models with non-trivial performance.** We compare the standard every 2nd layer initialization to models where we keep every 3rd, 4th, and 5th teacher layer when initializing the student. Every 3rd layer initialization produces similar interpolation behavior to every 2nd layer, but the smaller models do not interpolate smoothly, likely due to low student model performance and gaps in cosine similarity (see Appendix J).

**Results.**    In Figure 8, we find that student models initialized with every 2nd and every 3rd layer have similar interpolation performance, while the two smallest models do not have smooth interpolation behavior, which suggests that boomerang distillation works well when the student model shows non-trivial performance. We show in Appendix J that the cosine similarity between the output activations of the patched teacher block and the student layer it is replacing is correlated with interpolation performance. For instance, the drop in accuracy in every 4th layer after 3B parameters is primarily due to low cosine similarity between the patched teacher block and the student layer. This suggests that patching layers with high cosine similarity is a possible heuristic to consider for model interpolation to prevent a significant drop in performance.

### E.2    FURTHER COMPRESSING THE DISTILLED MODEL

**Setup.**    To test our hypothesis that the small models do not have clean interpolation behavior because of low cosine similarity, we initialize a set of small student models by manually selecting teacher layers to copy in a way that preserves alignment. To do so, we first compute the pairwise activation cosine similarity between each the output activations of each pair of teacher layers. We calculate the cosine similarity using a calibration dataset of 128 samples from The Pile (Gao et al., 2021). We then initialize the student by choosing teacher layers such that the first and last layer in each teacher block maintain high cosine similarity.

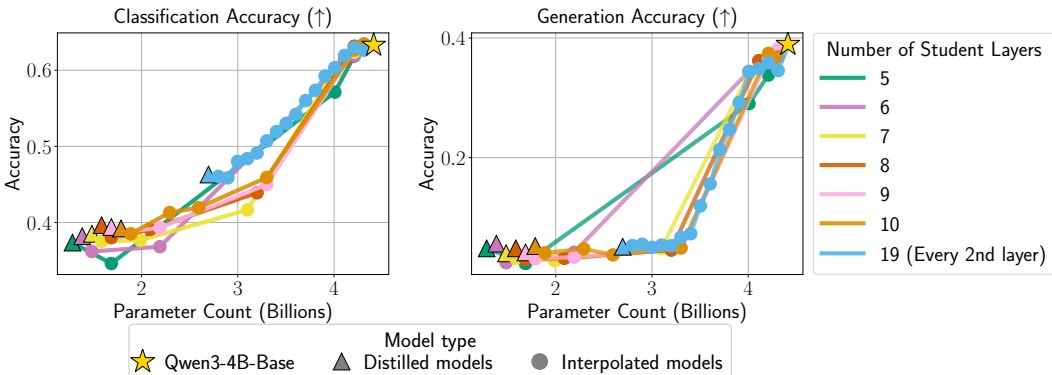

Figure 9: **Boomerang distillation occurs for very small student models with cosine similarity-informed initialization** Student models of size as small as 505M parameters have relatively smooth interpolation behavior when choosing the student initialization manually to preserve alignment. This indicates that the Qwen3-4B-Base teacher model can be compressed up to 8.7x if the initialization is performed in a way that preserves cosine similarity between the first and last layers in each teacher block.

**Results.** Figure 9 shows that by initializing the student in a way such that the teacher blocks have high similarity between the first and last layer, we can compress the student model up to 8.7x while preserving interpolation behavior. This validates our hypothesis from Appendix E.1 that low cosine similarity is the reason for poor interpolation performance for the naively initialized models in Figure 8. Although the student models can be compressed significantly beyond every 2nd layer, we note that these smaller models inherently produce a less granular set of interpolated models, since each student layer must be patched with its entire corresponding teacher block at once, allowing only for a maximum of $M - 1$ intermediate models (where $M$ is the number of student layers).

### E.3 IMPACT OF TRAINING TOKENS

In this experiment, we study the impact of training tokens and the performance of the interpolated models.

**Setup.** Following the same experimental setup from Section B, we train student models on different token budgets: 0.5B, 1B, 1.5B, 2B, 2.5B, and 3.1B. Depending on the token budget, we adjust the number of training steps and train the model for one epoch.

**Results.** We find that training the student models with more tokens results in better student models, which in turn creates better interpolated models (Figure 10). We also observe that if the distilled student model shows trivial performance (when trained with 0.5B tokens), the interpolated models also show trivial performance up to 4B parameters. In summary, for boomerang distillation to be effective, the student needs to show non-trivial performance.

## F THE BOOMERANG DISTILLATION PHENOMENON WITH QWEN, PYTHIA, AND LLAMA MODELS

In this section, we show boomerang distillation with Qwen3-8B-Base, Pythia-2.8B, Pythia-6.9B, and Llama-3.2-3B.

### F.1 QWEN3-8B-BASE AND QWEN3-14B-BASE

Figures 11 and 12 show that boomerang distillation occurs in the Qwen3-8B-Base and Qwen3-14B-Base models. Similar to Qwen3-4B-Base (§3.1), we observe a clear trend in performance as the size of the interpolated models increases. We also note that the student model created with Qwen3-8B-

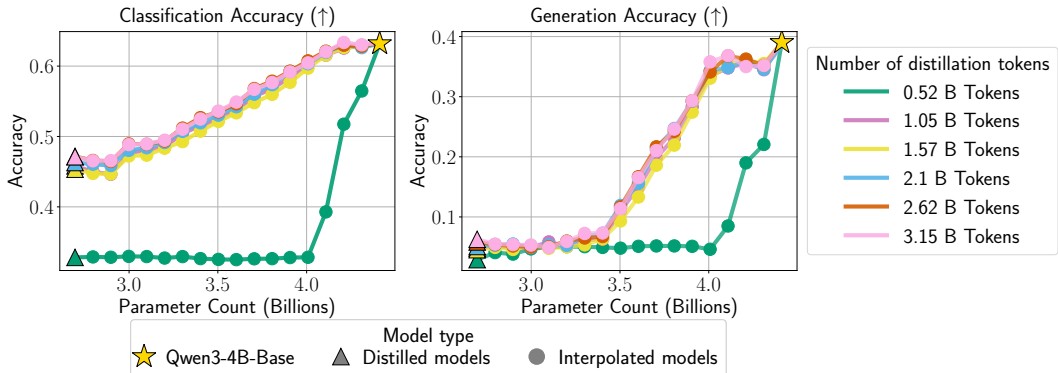

Figure 10: **Increased training token budget produces better interpolated models.** Distilling the student model on more tokens results in distilled models with higher performance, which create better interpolated models. Distilled student models with trivial performance (0.52B tokens) do not have smooth interpolation behavior, indicating that non-trivial student model performance is necessary for boomerang distillation to occur.

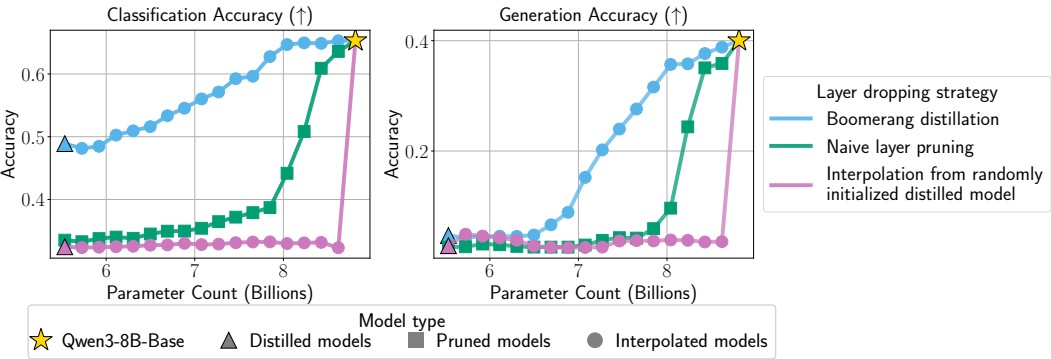

Figure 11: **Boomerang distillation with Qwen 8B creates models with smoothly interpolated size and performance.**

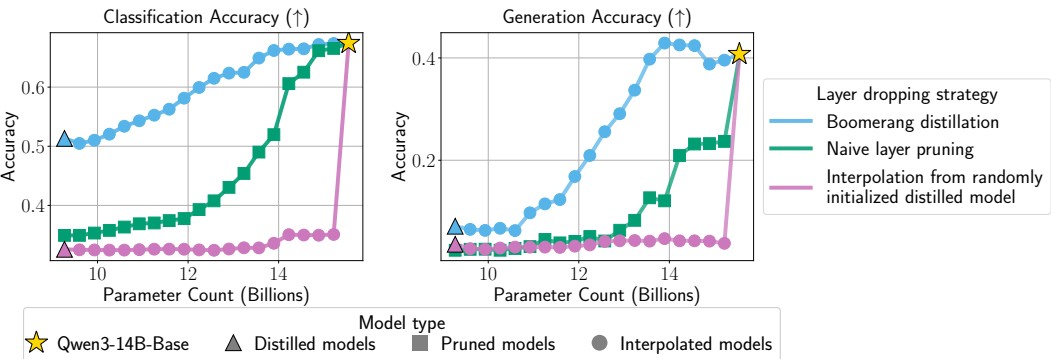

Figure 12: **Boomerang distillation with Qwen 14B creates models with smoothly interpolated size and performance.**

Base is approximately 5.6B parameters in size, which is close to the pretrained Qwen-3-4B-Base model but performs significantly worse. Similarly, the student created with Qwen3-14B-Base has around 8.5B parameters and performs worse than Qwen3-8B-Base. We suspect that the corpus used to pretrain Qwen is of a higher quality and trained on significantly more tokens compared to the distilled student model, which leads to improved out-of-the-box performance. In such cases, for a given size, we recommend choosing the model interpolated from the closest pretrained model.

### F.2 PYTHIA-2.8B, PYTHIA-6.9B, AND PYTHIA 12B

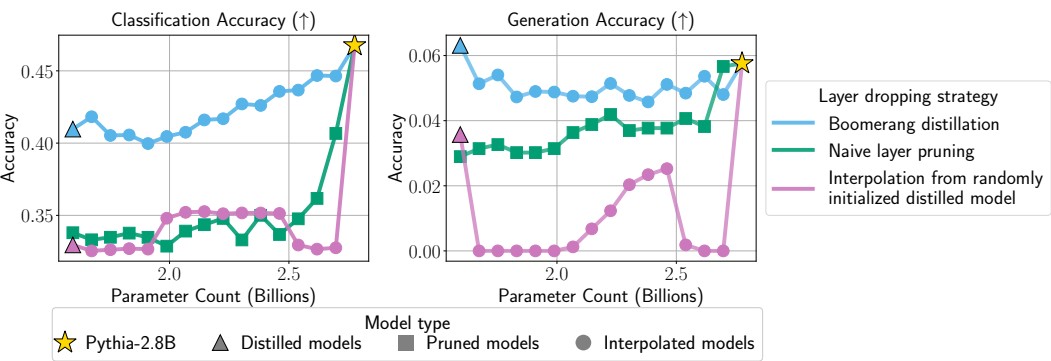

Figure 13: **Boomerang distillation with Pythia 2.8B creates models with smoothly interpolated size and performance.**

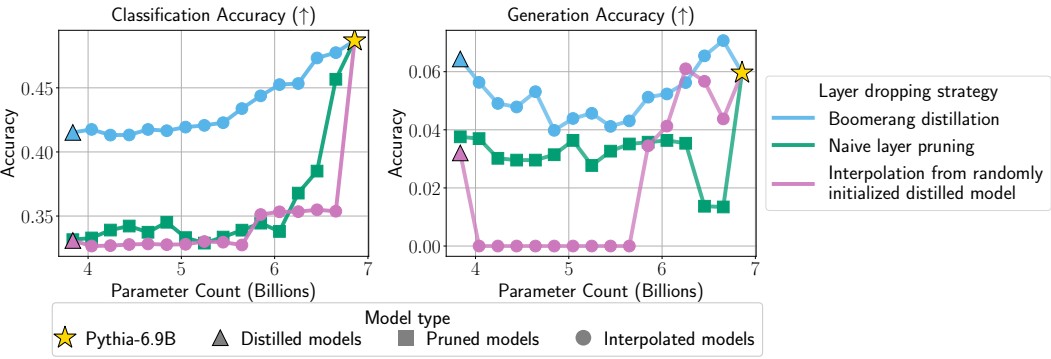

Figure 14: **Boomerang distillation with Pythia 6.9B creates models with smoothly interpolated size and performance.**

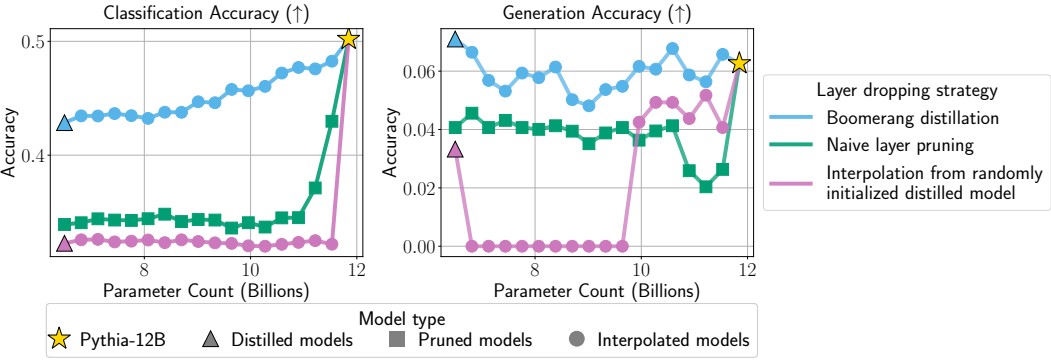

Figure 15: **Boomerang distillation with Pythia 12B creates models with smoothly interpolated size and performance.**

Figures 13, 14, and 15 show boomerang distillation with Pythia 2.8B, Pythia 6.9B, and Pythia 12B models. In all cases, we see that interpolated shows improved performance in classification accuracy, but their performance on generation tasks is nearly 0%. We also observe that the performance of the pretrained models is close to 0%, which suggests that boomerang distillation may not improve the performance of the interpolated models beyond the performance of the pretrained models.

### F.3 LLAMA-3.2-3B

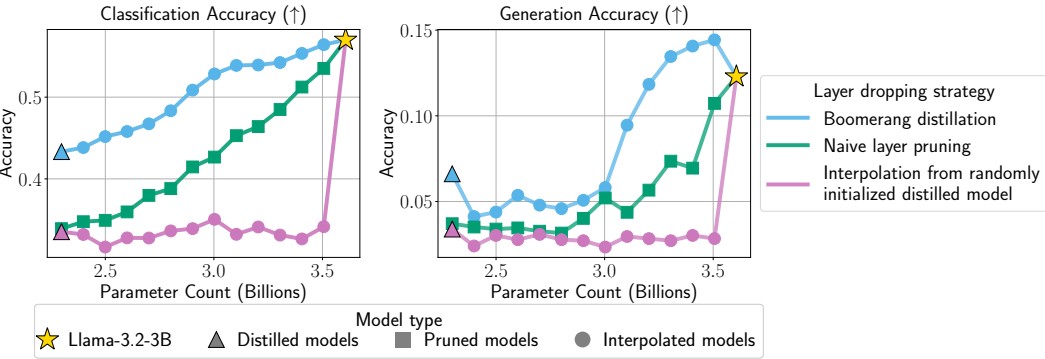

Figure 16: **Boomerang distillation with Llama-3.2-3B creates models with smoothly interpolated size and performance.**

Figure 16 shows the boomerang distillation phenomenon in Llama-3.2-3B. We modify the initialization and student patching order setup due to the behavior of the first base model layer to ensure that first-layer information is preserved (see Appendix I for details). We find that boomerang distillation with Llama-3.2-3B as a base model produces interpolated models with smoothly interpolated performance across classification and generation tasks. In contrast, naive layer pruning and interpolation using a randomly initialized distilled student model do not recover smoothly interpolated models.

## G PYTHIA-2.8B FULL RESULTS

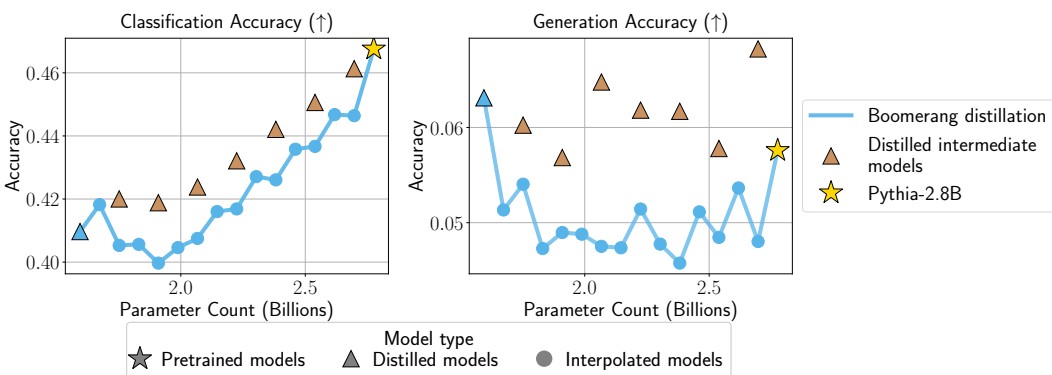

Figure 17: **Interpolated models produced using boomerang distillation and Pythia-2.8B have comparable performance to pretrained and naively distilled models.**

**Comparison to Standard Knowledge Distillation.** Figure 17 shows that interpolated models created using boomerang distillation for Pythia-2.8B have comparable performance to the intermediate models trained using standard knowledge distillation. Unlike the Qwen models, the models trained with standard distillation perform better than interpolated models across all sizes, suggesting that Qwen models are trained on a much higher quality corpus, and training them on The Pile drops their performance.

**Effect of Knowledge Distillation.** Figure 18 shows that student models trained with a cross entropy and an alignment loss create interpolated models with lower perplexity. We also observe that the interpolated models incorporating cross entropy and alignment losses show meaningful differences in classification accuracy compared to models trained without them, particularly at smaller

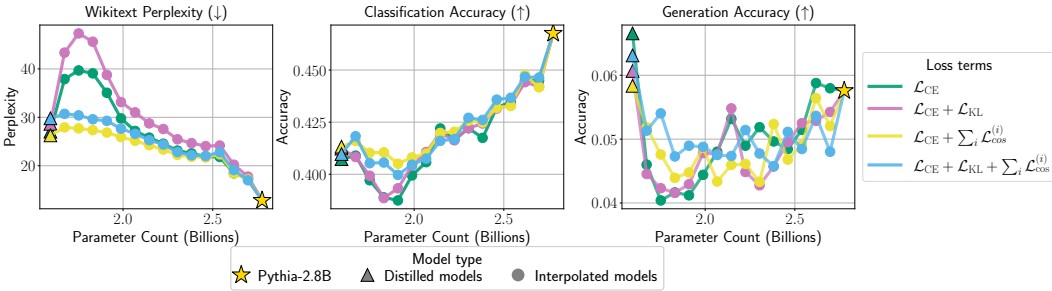

Figure 18: **Per-layer loss yields stable and smoother interpolation performance in Pythia-2.8B.**

model sizes. Finally, we see that the interpolated models show trivial performance on generation tasks since the teacher performs poorly on those tasks.

**Comparison to Layer Pruning Methods.**    Figure 19 shows that boomerang distillation and layer pruning exhibit similar performance on the classification tasks. While boomerang distillation shows stronger performance at smaller model sizes, we see that LaCo and ShortGPT show stronger performance at larger model sizes. Since the pretrained teacher model itself does not show strong performance, we suspect the patching order makes a difference in performance. Nevertheless, boomerang distillation is competitive with existing approaches in the layer pruning.

## H    LLAMA-3.2-3B FULL RESULTS

**Comparison to Standard Knowledge Distillation.**    Figure 20 shows that boomerang distillation creates interpolated models that show comparable performance to the models trained with standard knowledge distillation, and even outperforms them at larger sizes. Similar to Qwen, since we do not have access to the Llama's pretraining corpus, we find that distilling on The Pile leads to a performance drop at larger sizes. On the other hand, boomerang distillation retains the benefits of pretraining and outperforms standard distillation in some cases.

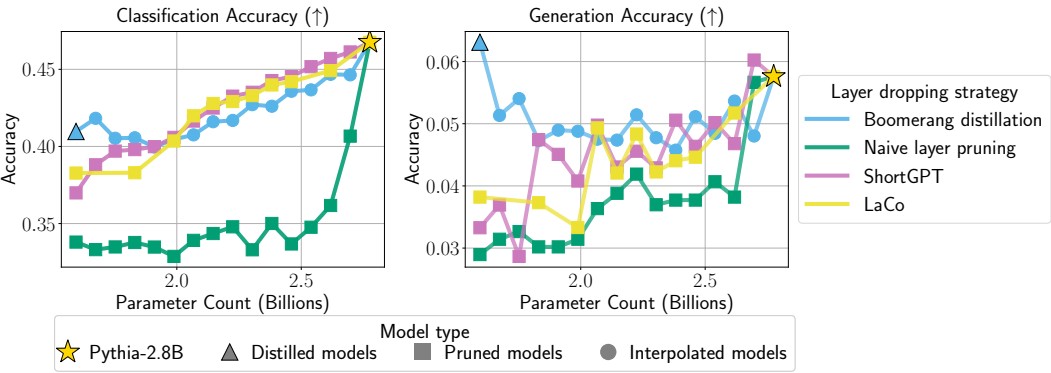

Figure 19: **Boomerang distillation with Pythia-2.8B performs similarly to depth pruning methods.**

**Effect of Knowledge Distillation.**    Figure 21 shows that training with alignment loss, i.e., cosine distance loss, creates interpolated models with lower perplexity across most intermediate sizes. The classification accuracy is also slightly higher, especially for models with around 2.5-3B inference parameters. Similarly to the Qwen models, we see that training without alignment loss degrades generation performance at high parameter counts, likely due to the importance of the last layers.

**Comparison to Layer Pruning Methods.**    Figure 22 shows that boomerang distillation outperforms layer pruning approaches at all sizes. We observe that the gap in classification accuracy is

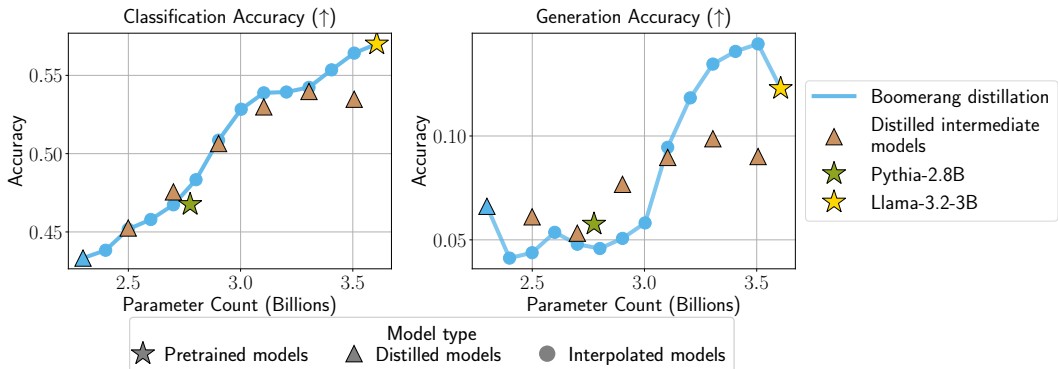

Figure 20: **Interpolated models produced using boomerang distillation and Llama-3.2-3B have comparable performance to pretrained and naively distilled models.**

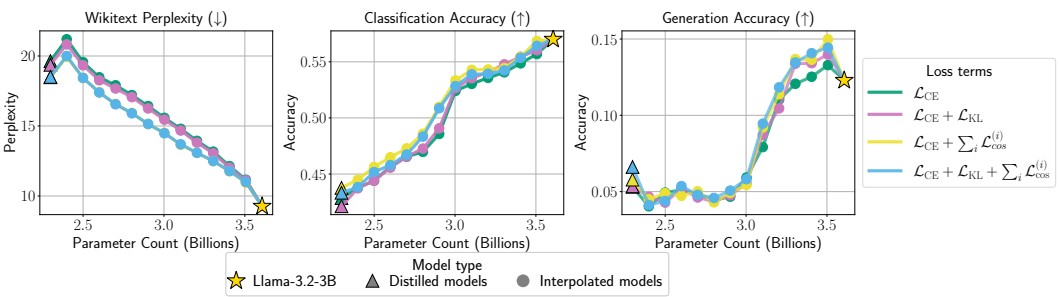

Figure 21: **Per-layer loss yields stable and smoother interpolation performance in Llama-3.2-3B.**

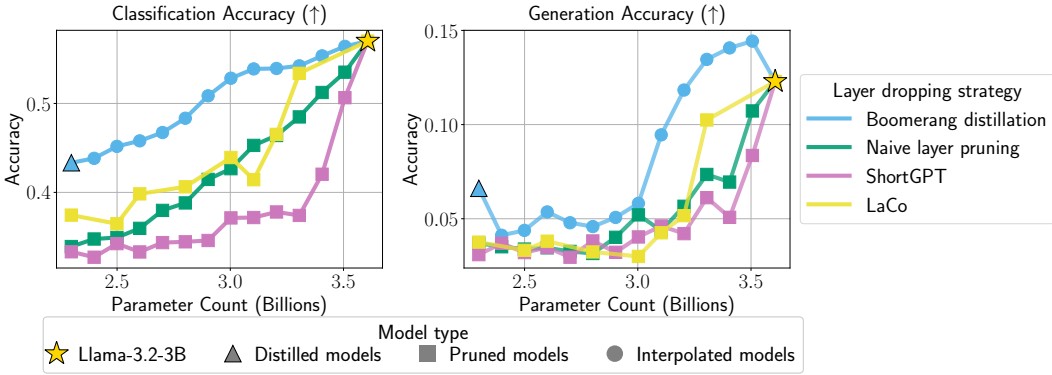

Figure 22: **Boomerang distillation with Llama-3.2-3B performs significantly better than depth pruning methods.**

initially quite high, but around 3.2B, LaCo recovers performance and performs competitively to boomerang distillation. These results suggest that the interpolated models created using boomerang distillation perform significantly better than existing approaches, but they perform similarly to LaCo as the model size approaches that of the pretrained teacher model.

## I  LLAMA-3.2-3B COSINE SIMILARITY ANALYSIS

In this section, we analyze the per-layer activation cosine similarity between the output activations of all pairs of distilled and base model layers. We compute the activations on a calibration set

consisting of 128 samples from The Pile (Gao et al., 2021) and report the mean cosine similarity across all tokens in each sample. We find that per-layer cosine similarity explains when boomerang distillation is noisy or has poor interpolation performance. Our results indicate that the best practices when using boomerang distillation are to (1) patch student layers with low cosine similarity first and (2) ensure that consecutive layers with low cosine similarity are not pruned when initializing the student model.

**Standard Initialization.** Figure 23 shows the cosine similarity analysis for the distilled model created by initializing the student model from Llama-3.2-3B by pruning every other layer and keeping the last two layers:

$$\boldsymbol{\theta}_S = (\boldsymbol{\theta}_T^E, \boldsymbol{\theta}_T^{(1)}, \boldsymbol{\theta}_T^{(3)}, \ldots, \boldsymbol{\theta}_T^{(N-3)}, \boldsymbol{\theta}_T^{(N-1)}, \boldsymbol{\theta}_T^{(N)}, \boldsymbol{\theta}_T^D) \tag{2}$$

Figure 23 demonstrates that the output activations of the first layer in the distilled student model have high cosine similarity to the output activations of the first layer in the teacher model, but have low cosine similarity to the outputs of the second layer in the teacher. As a result, the remaining layers in the distilled student do not have high cosine similarity to their corresponding base model layers until the last two layers of the student model. This means that the model does not recover smoothly interpolated performance when patching layers of the student model starting from the last layers of the model (Figure 24 green line). In Figure 24 (blue line), we show that this issue can be mitigated by patching from the first layers of the model. Thus, beginning the student patching process with layers with low cosine similarity to their corresponding teacher layers provides a way to improve interpolation performance.

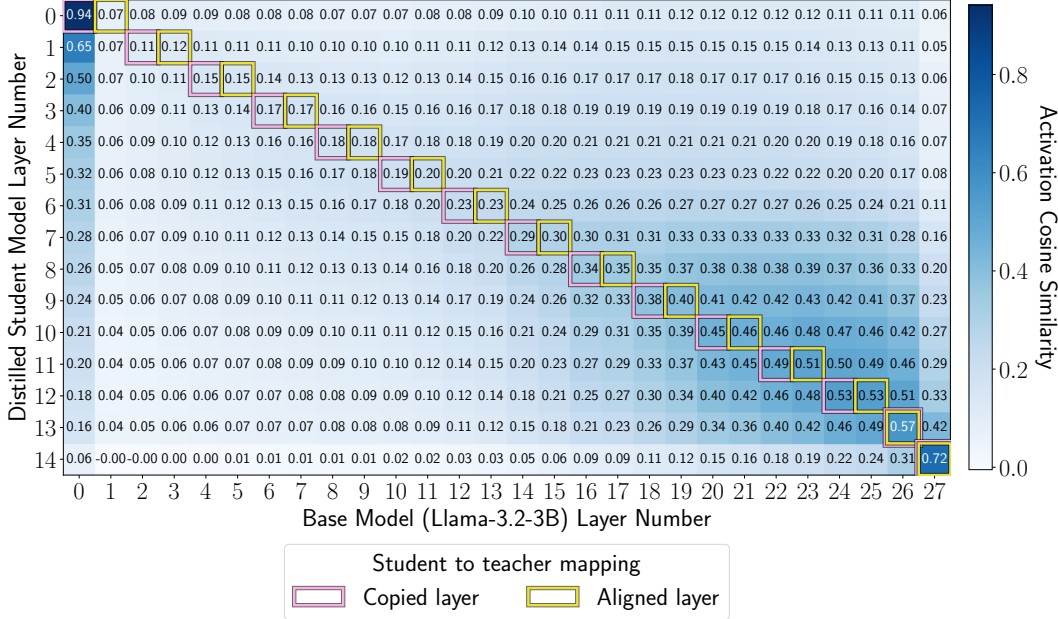

Figure 23: **Per-layer cosine similarity between the output activations of the distilled student model and the teacher model, Llama 3.2 3B.** The first student and teacher layer exhibit high cosine similarity, but all student layers except for the last one have low cosine similarity to their corresponding teacher block (layer $\boldsymbol{\theta}_T^{(\ell_i)}$ shown in pink and layer $\boldsymbol{\theta}_T^{(\ell_{i+1}-1)}$ shown in yellow). As a result, patching the student model starting from the last layer does not smoothly recover the interpolated performance (Figure 24).

**Preserving First-Layer Information.** In Figure 25, we consider an alternative student initialization to solve the misalignment in the first student layer (Figure 23), where we instead keep the first two teacher layers and alternate layers for the remaining initialization:

$$\boldsymbol{\theta}_S = (\boldsymbol{\theta}_T^E, \boldsymbol{\theta}_T^{(1)}, \boldsymbol{\theta}_T^{(2)}, \boldsymbol{\theta}_T^{(4)}, \ldots, \boldsymbol{\theta}_T^{(N-2)}, \boldsymbol{\theta}_T^{(N)}, \boldsymbol{\theta}_T^D) \tag{3}$$

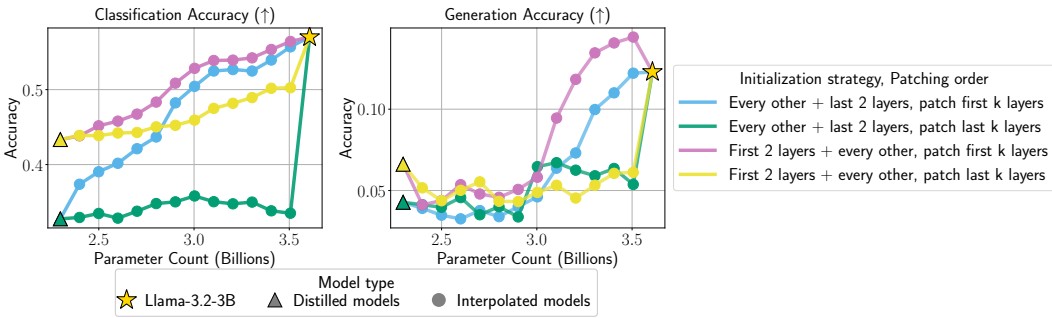

Figure 24: **Model size interpolation with different student initialization and patching order with Llama 3.2 3B.** We find that the distilled student model trained by initializing with the first two layers and every other layer from the teacher model, and then patching from the first layer to the last, creates the best interpolated models in Llama 3.2 3B.

We choose this initialization because we hypothesize that given the low cosine similarity between the first and second layers in the model (Figure 23), combining the first two base model layers into one student layer needlessly decreases the alignment between subsequent student and base model layers. In Figure 25, we find that keeping the first and second Llama-3.2-3B layers indeed results in significantly higher cosine similarity between student and base model layers. This translates to significantly higher student and interpolation performance (Figure 24 pink and yellow lines), especially when combined with our strategy of patching layers starting from the first model layers (Figure 24 pink line).

**Takeaways.** In summary, we observe that for some base models (such as Llama-3.2-3B), naive initialization and patching approaches are insufficient. We identify low cosine similarity between key base model layers as a contributing factor to this issue. We find that we can improve performance by choosing a student patching order that prioritizes blocks of teacher layers with low cosine similarity or by initializing the student model in a manner that ensures the activations of the first and last layer in each block $\mathbf{b}^{(i)}$ do not have low cosine similarity to each other.

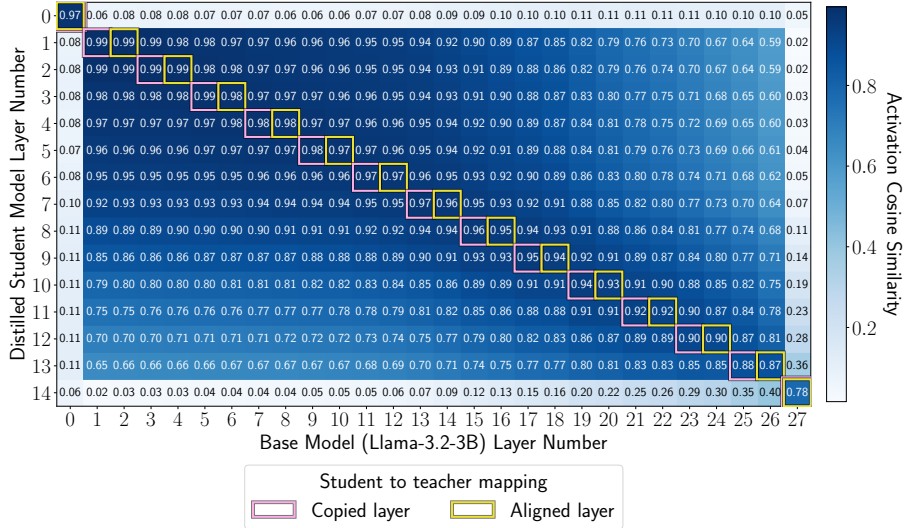

Figure 25: **Per-layer cosine similarity between the output activations of the distilled student model initialized with the first two teacher layers and the teacher model, Llama 3.2 3B.** After distilling a student model initialized with the first two layers (Equation 3), all student layers have high cosine similarity to their corresponding teacher block (layer $\boldsymbol{\theta}_T^{(\ell_i)}$ shown in pink and layer $\boldsymbol{\theta}_T^{(\ell_{i+1}-1)}$ shown in yellow). Thus, student patching recovers smooth interpolation (Figure 24).

## J    STUDENT MODEL SIZE ABLATION COSINE SIMILARITY ANALYSIS

Here, we study the per-layer cosine similarity between each pair of distilled and base model layers in the student model initialized with every 4th layer in Figure 8. We use the same setup as in §I to calculate the cosine similarity values. Figure 26 shows that student layers 0 and 4-11 have high cosine similarity to the outputs of their corresponding teacher blocks (shown in yellow). In contrast, student layers 2-3 have low cosine similarity to the input and output activations of their corresponding teacher blocks, while layer 1 has high cosine similarity to the first layer in its teacher block but not the last one. Thus, when student layers are patched starting from the last layers of the model in Figure 8, the performance increases for the first 4 patched student layers $(7, 6, 5, 4)$, then decreases in the low cosine similarity region when patching $(3, 2)$ before increasing again. This supports the results in §I and indicates that cosine similarity between the student layer and the layers in its corresponding teacher block is correlated with interpolation performance.

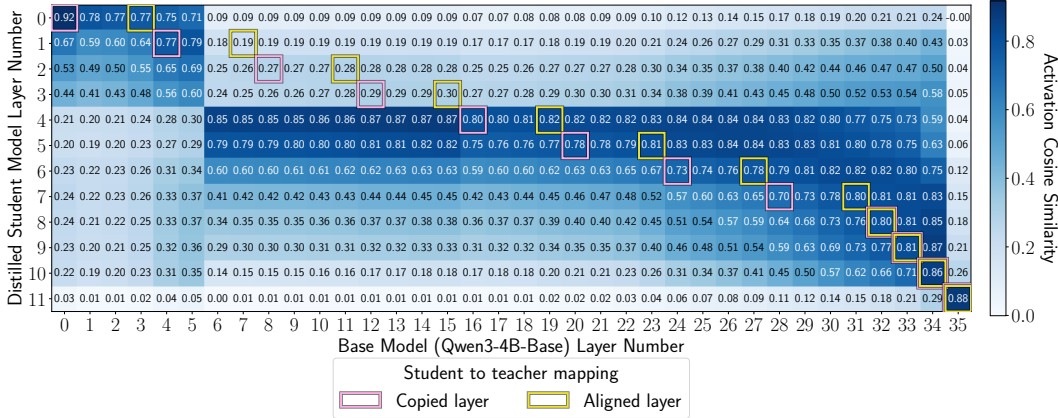

Figure 26: **Per-layer cosine similarity between the output activations of the distilled student model initialized with every 4th layer and the teacher model, Qwen3-4B-Base.** Student layers with high cosine similarity to the outputs of their teacher blocks have predictable interpolation performance when patched in Figure 8. On the other hand, student layers with low cosine similarity see a decrease in interpolation performance when they are patched.

## K    WHY DOES BOOMERANG DISTILLATION WORK?

In this section, we provide an intuition and proof-of-concept experiment to demonstrate why boomerang distillation works.

**High-Level Intuition.**    The main idea behind boomerang distillation is that we ensure through aligned initialization and distillation that not only does the student approximate the teacher model, but each student layer approximates the functionality of its corresponding teacher block. Then, patching a student layer with its corresponding teacher block produces a better approximation of the teacher model.

**Proof-of-Concept Experiment.**    To show that patching a higher number of student layers creates intermediate models that better approximate the teacher, we compute the last layer cosine similarity between the activations of the interpolated and the teacher model for each number of layers patched. We use a held-out calibration set of 128 texts from The Pile (Gao et al., 2021) to compute the cosine similarity. Figure 27 demonstrates that across different models, the last layer cosine similarity between the interpolated and teacher models increases as the number of layers patched increases. This confirms our intuition that patching more layers produces interpolated models that better approximate the teacher.

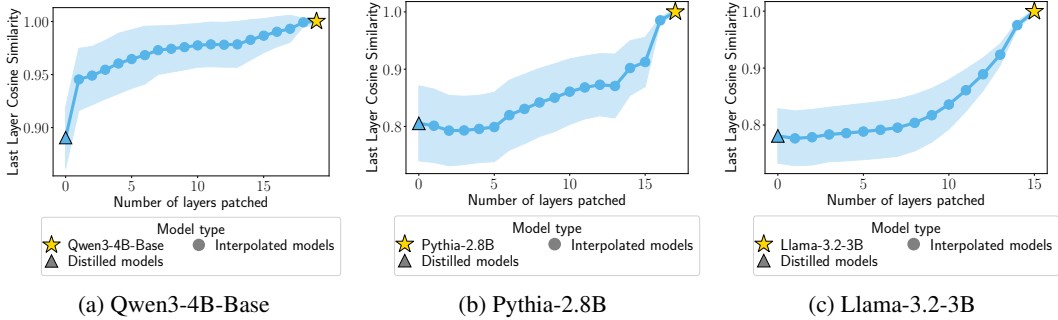

(a) Qwen3-4B-Base   (b) Pythia-2.8B   (c) Llama-3.2-3B

Figure 27: **Last layer cosine similarity between interpolated and teacher models increases as more layers are patched acrosss models.** As more student layers are patched, the interpolated models better approximate the teacher model.

## L    COMPUTATIONAL COMPLEXITY

In this section, we compute the floating point operations per second (FLOPS) for boomerang distillation versus standard distillation. We compute FLOPS for 10 iterations during training using the `flops-profiler` package (Li, 2023) and report the training cost computed by multiplying the average FLOPS per iteration by the total number of training iterations in Table 3. Boomerang distillation is an instance of amortized training cost, as after the initial distillation training, boomerang distillation can be used to obtain intermediate models in a zero-shot manner. As a result, boomerang distillation has equivalent computational compexity to distilling a single model of student size, and requires 14.53-19.17x less FLOPS compared to training each intermediate model independently.

| Teacher Model | Models | FLOPS | Theoretical Compute Speedup |
|---|---|---|---|
| Qwen3-4B-Base | Standard distillation | 8.27e20 | - |
|  | Boomerang distillation | 4.31e19 | 19.17x |
| Pythia-2.8B | Standard distillation | 4.77e20 | - |
|  | Boomerang distillation | 2.80e19 | 17.01x |
| Llama-3.2-3B | Standard distillation | 5.07e20 | - |
|  | Boomerang distillation | 3.49e19 | 14.53x |

Table 3: **Boomerang distillation provides significant computational speedup compared to individually distilling intermediate models.** For Qwen3-4B-Base, Pythia-2.8B, and Llama-3.2-3B, we report the FLOPS required to individually distill each intermediate model versus boomerang distillation for the same number of training tokens (2.1B tokens). We can reduce FLOPs by 19.17x for Qwen, 17.01x for Pythia, and 14.53x for Llama using boomerang distillation.

## M    ADDITIONAL EVALUATION RESULTS

Here, we provide perplexity, per-task classification accuracy, and per-task exact match generation accuracy for experiments in §3.

### M.1    PERPLEXITY

**Comparison to Naive Layer Pruning and Random Initialization.** Figure 28 shows that boomerang distillation interpolates smoothly in terms of perplexity between the student and the distilled model, while perplexity degrades for naive layer pruning as more layers are dropped. All models interpolated from a randomly initialized distilled model have a perplexity above $10^4$.

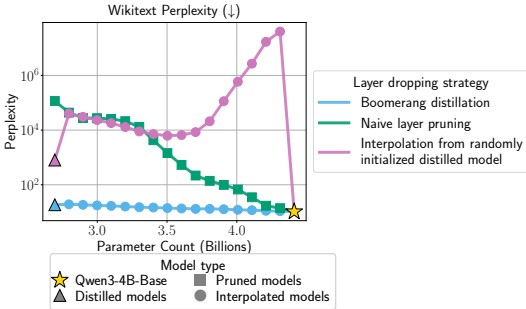

Figure 28: **Boomerang distillation creates models with smoothly interpolated size and performance.**

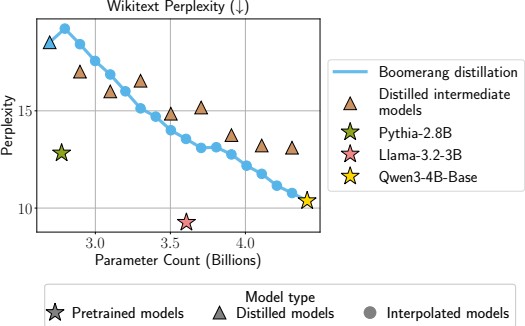

Figure 29: **Interpolated models produced using boomerang distillation have comparable performance to pretrained and naively distilled models.**

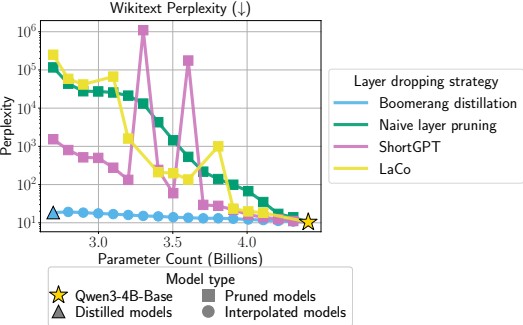

Figure 30: **Boomerang distillation performs significantly better than depth pruning methods.**

**Comparison to Standard Knowledge Distillation.** In Figure 29, small distilled models have slightly lower perplexity than interpolated models, while larger distilled models have slightly higher perplexity than interpolated models. This follows our observations in §3.2. However, one notable difference is that while pretrained Pythia-2.8B and Llama-3.2-3B models have similar classification and generation performance but lower perplexity than interpolated models. This is likely due to their extensive pretraining on next-token prediction.

**Comparison to Layer Pruning Methods.** In Figure 30, we show that boomerang distillation interpolates smoothly in terms of perplexity between the student and the distilled model, while all layer pruning approaches increase significantly in perplexity after more than six layers are dropped.

## M.2 CLASSIFICATION TASKS

In this section, we report per-task classification accuracy in Figures 31-34 for experiments in §3. We find that the per-task results for all experiments align with the mean performance reported in §3.

## M.3 GENERATION TASKS

Here, we show per-task generation exact match accuracy in Figures 35-38 for experiments in §3. We find similar trends in per-task generation performance as reported for the mean generation accuracy in §3.

# N PRUNING METHOD DETAILS

In this section, we describe how we prune layers in the comparisons to Layer Collapse (LaCo) (Yang et al., 2024) and ShortGPT (Men et al., 2024) in Figures 7, 19, 22, 34, and 38.

**LaCo.** LaCo loops through all the model layers and iteratively merges chunks of $\mathcal{C}$ layers if the cosine similarity of the last layer hidden activations of the merged model and the last layer hidden activations of the original model is above a certain threshold $\mathcal{T}$. The LaCo layer merging operation for a chunk starting at layer $\ell$ is performed by adding the difference in weights between each merged layer and $\boldsymbol{\theta}^{(\ell)}$ to $\boldsymbol{\theta}^{(\ell)}$ to create a new model $\boldsymbol{\theta}^*$, where

$$\boldsymbol{\theta}^{*(\ell)} = \boldsymbol{\theta}^{(\ell)} + \sum_{i=1}^{\mathcal{C}} \boldsymbol{\theta}^{(\ell+i)} - \boldsymbol{\theta}^{(\ell)} \tag{4}$$

In order to construct the LaCo models, we compute the cosine similarity values on a held-out calibration set of 16 samples from the Pile (Gao et al., 2021). We follow the hyperparameter setup detailed in the appendix of Yang et al. (2024). We set the layer range parameters $\mathcal{L} = 1$ and $\mathcal{H} = N$, where $N$ is the number of teacher layers. We also fix the minimum interval parameter $\mathcal{I} = 2$. To generate models with different numbers of layers, we sweep over the set of threshold values $\mathcal{T}$ and number of layers merged per operation $\mathcal{C}$ included in Yang et al. (2024). We provide the hyperparameter details in Table 4.

| LaCo Hyperparameters | Values |
|---|---|
| Number of layers merged per operation ($\mathcal{C}$) | $\{3, 4, 5, 6\}$ |
| Start of layer range ($\mathcal{L}$) | 1 |
| End of layer range ($\mathcal{H}$) | Number of teacher layers $N$ |
| Minimum interval ($\mathcal{I}$) | 2 |
| Threshold ($\mathcal{T}$) | $\{0.95, 0.85, 0.75, 0.65, 0.55, 0.45\}$ |

Table 4: Hyperparameters used to create LaCo models in Figures 7, 19, 22, 34, and 38

**ShortGPT.** In ShortGPT, model layers are pruned by first computing the Block Importance (BI) score, or the cosine distance between the input and output activations for the layer:

$$\text{BI}^{(i)} = 1 - \mathbb{E}_{X,j} \left[ \frac{\boldsymbol{x}_j^{(i)} \cdot \boldsymbol{x}_j^{(i+1)}}{||\boldsymbol{x}_j^{(i)}|| \, ||\boldsymbol{x}_j^{(i+1)}||} \right]$$

Then, layers are removed sequentially by pruning the layer with the lowest BI score. We compute BI with respect to a held-out set of 128 calibration texts from the Pile (Gao et al., 2021).

# O USE OF LARGE LANGUAGE MODELS

We utilized generative AI tools for code completion, debugging, and minor grammatical corrections in the manuscript. The authors carried out all the substantive research contributions, analyses, and interpretations.

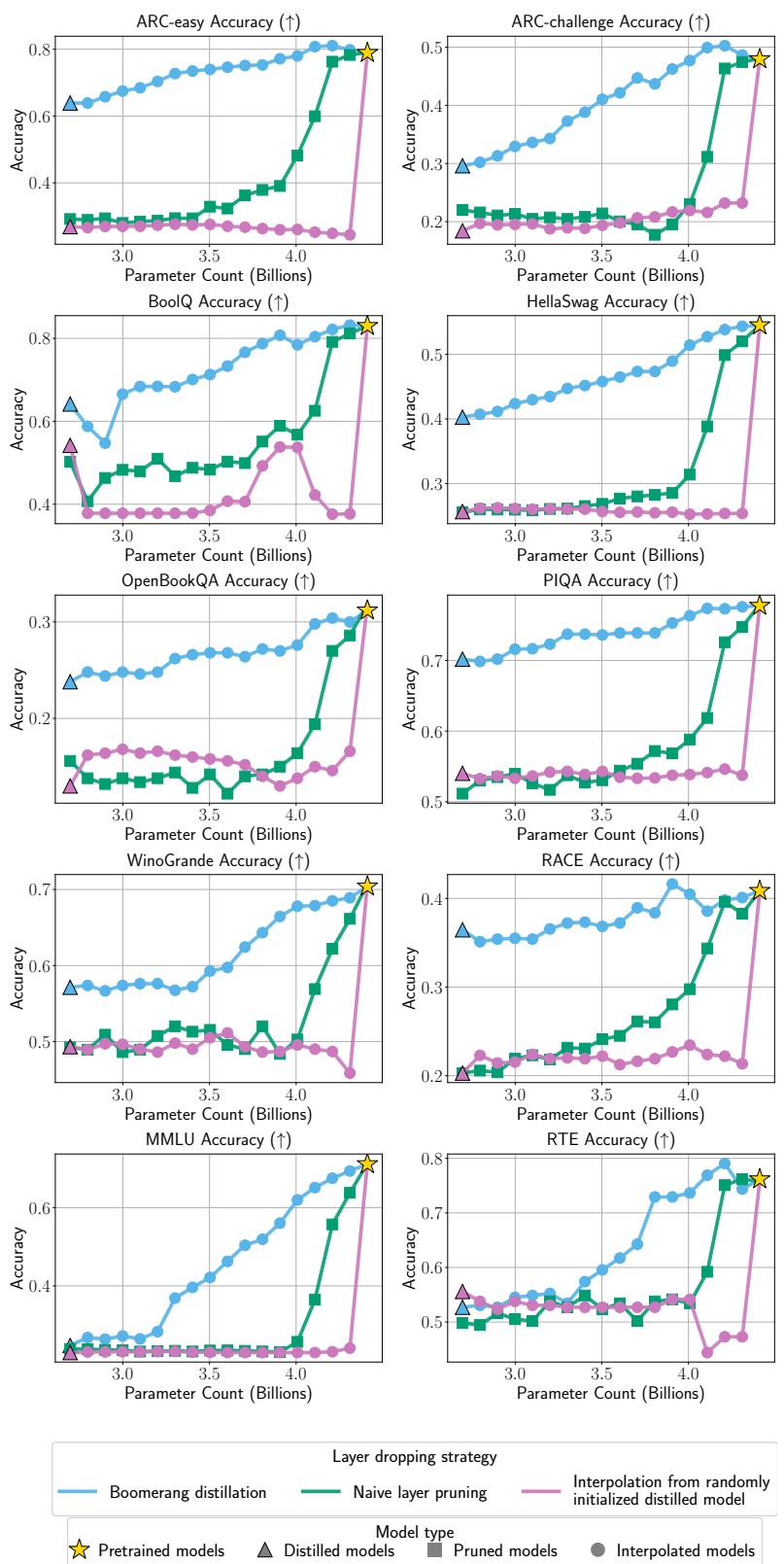

Figure 31: **Boomerang distillation creates models with smoothly interpolated size and per-task classification accuracy.**

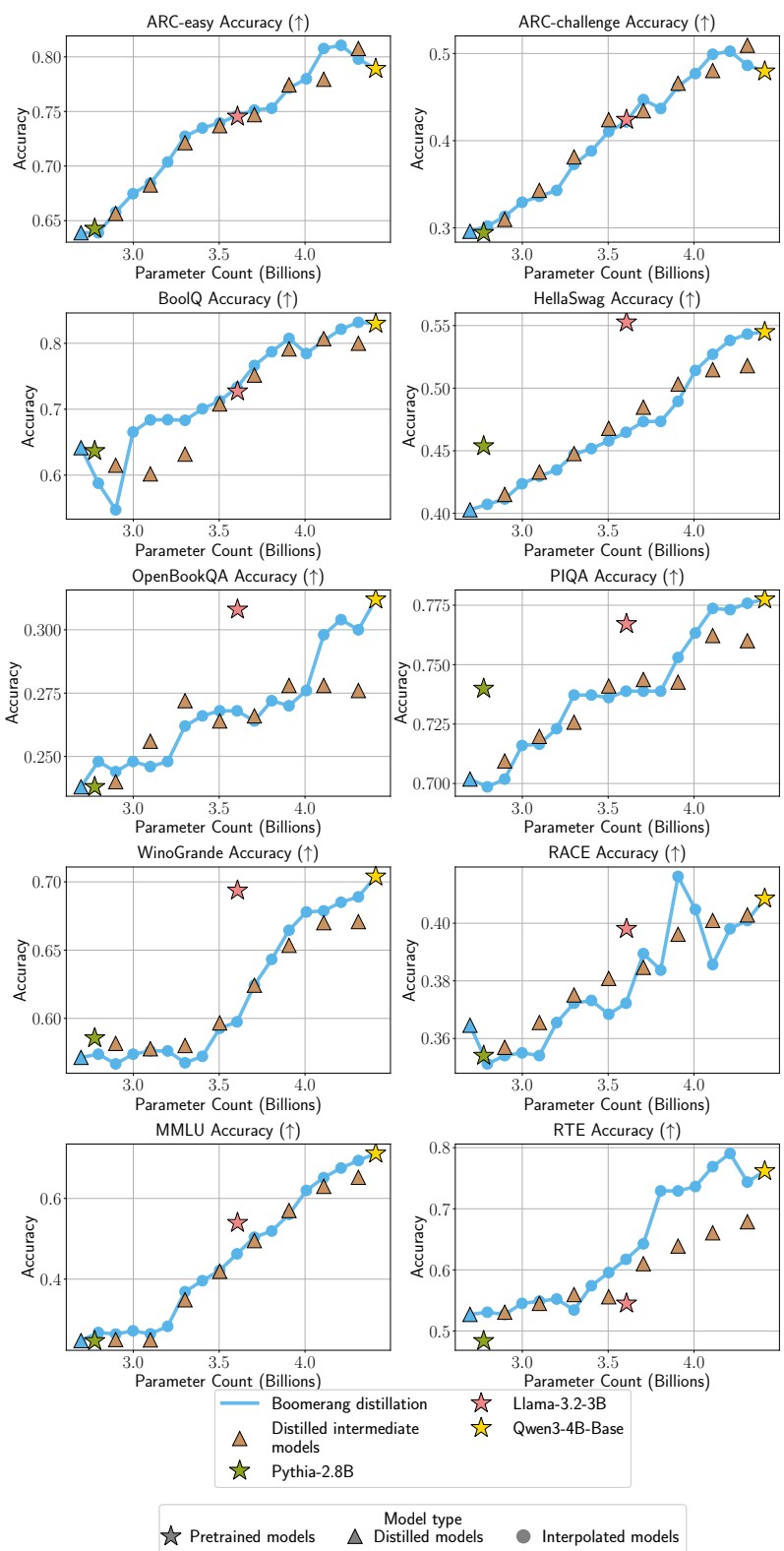

Figure 32: **Interpolated models produced using boomerang distillation have comparable per-task classification accuracy to pretrained and naively distilled models.**

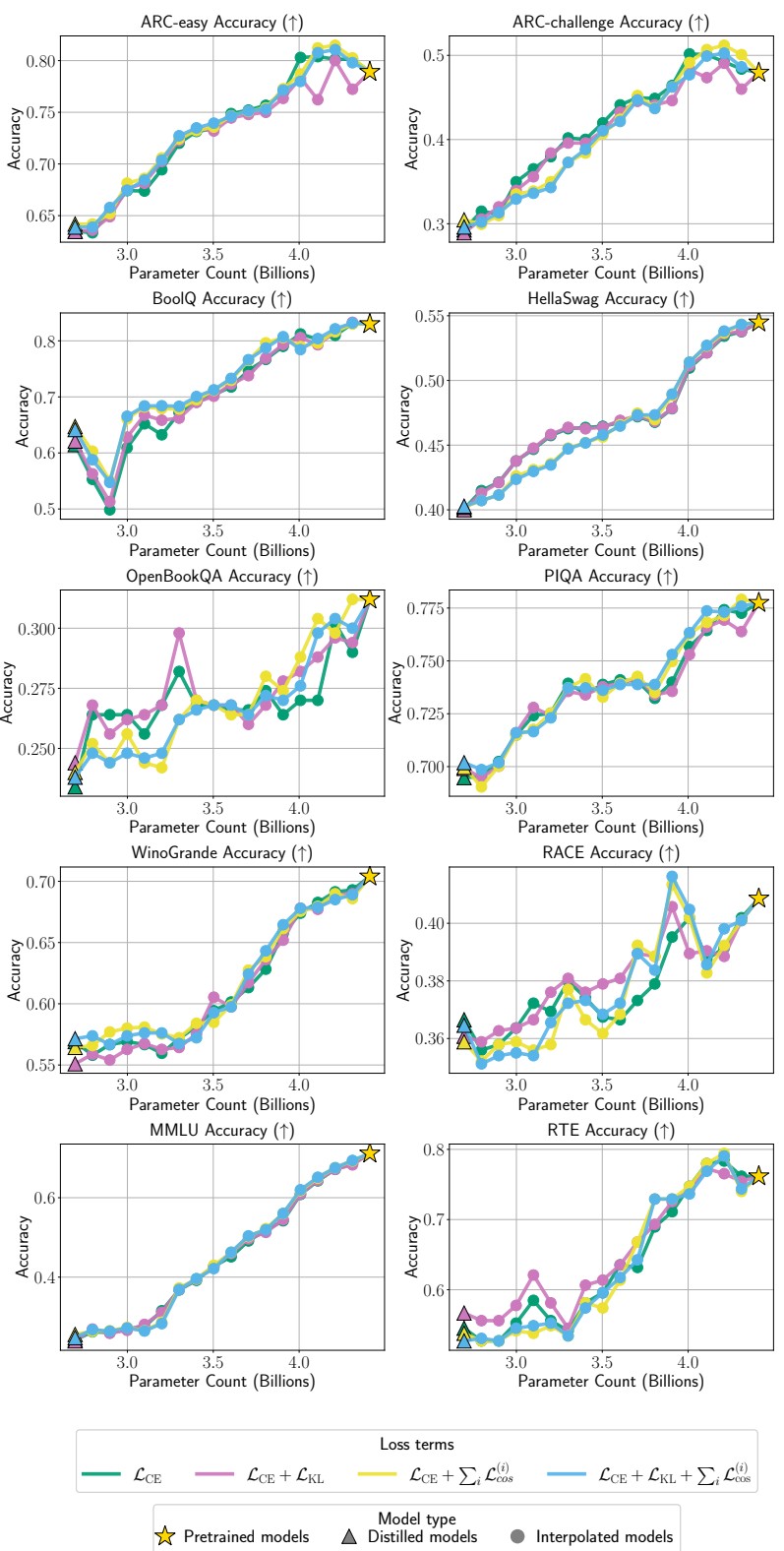

Figure 33: **Per-layer loss yields stable and smoother per-task classification accuracy for interpolated models.**

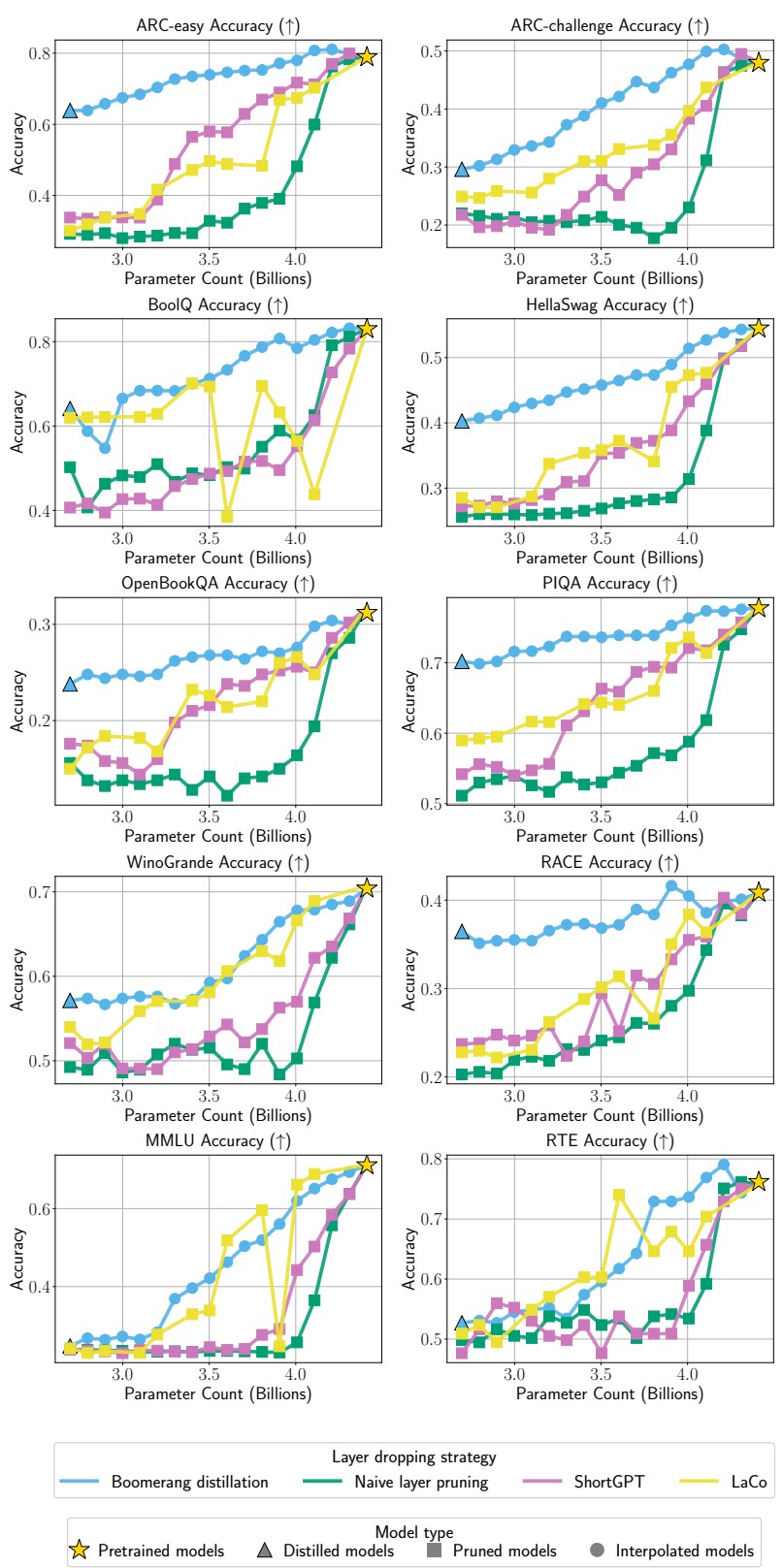

Figure 34: **Boomerang distillation has significantly better per-task classification accuracy than depth pruning methods.**

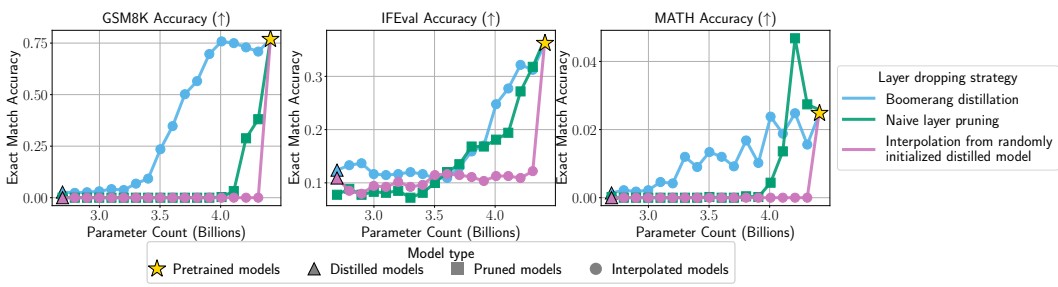

Figure 35: **Boomerang distillation creates models with smoothly interpolated size and per-task generation accuracy.**

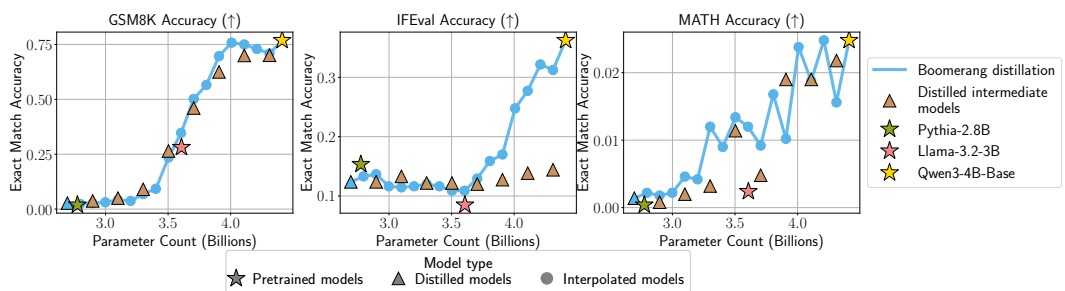

Figure 36: **Interpolated models produced using boomerang distillation have comparable per-task generation accuracy to pretrained and naively distilled models.**

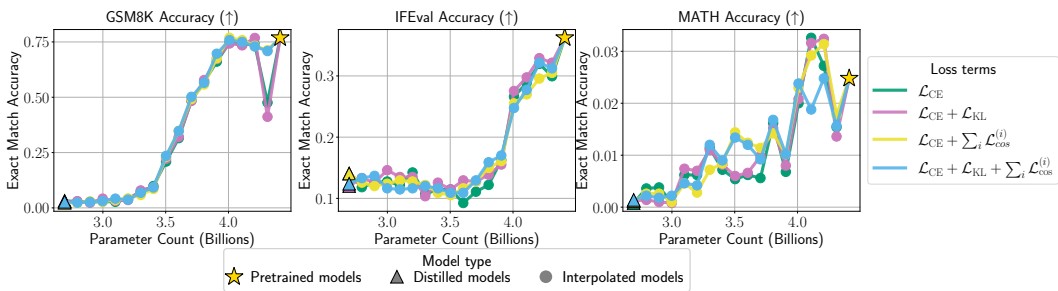

Figure 37: **Per-layer loss yields stable and smoother per-task generation accuracy for interpolated models.**

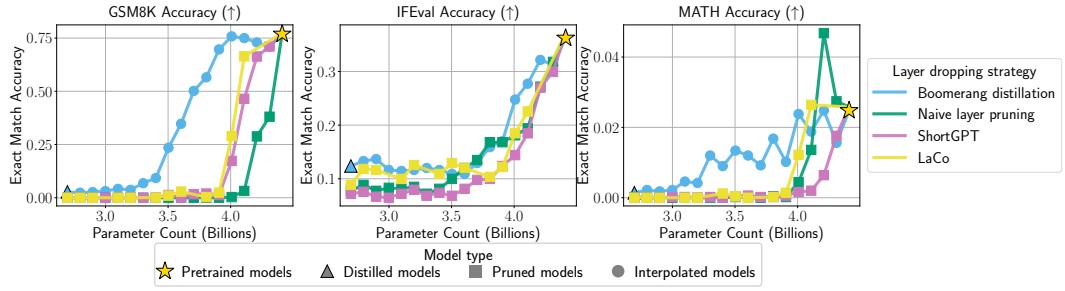

Figure 38: **Boomerang distillation has significantly better per-task generation accuracy than depth pruning methods.**

