# OpenReview forum: "Boomerang Distillation Enables Zero-Shot Model Size Interpolation"
_ICLR.cc/2026/Conference — ICLR 2026 Poster_

### Official Review · Reviewer_Cce4 · 2025-10-24

**Soundness:** 3
**Presentation:** 4
**Contribution:** 4
**Rating:** 8
**Confidence:** 2

**Summary:**

This paper introduces and empirically validates a novel phenomenon termed "boomerang distillation." The method provides a highly efficient, zero-shot technique for creating a family of models with fine-grained sizes that interpolate between a small, distilled "student" model and a large "teacher" model. The authors conduct extensive experiments across multiple model families (Qwen, Pythia, Llama) and sizes, demonstrating that this method produces models whose performance scales smoothly with size. These interpolated models consistently and significantly outperform naive layer pruning and often match or even surpass the performance of models of equivalent size that are trained individually via standard distillation, all while dramatically reducing the computational cost of creating model families.

**Strengths:**

- This article proposes a simple, three-stage scheme for zero-shot model size interpolation, which is both novel and highly practical. The ability to generate a series of model sizes after just one training run significantly reduces the computational resources required to build and deploy model families.
- The paper’s claims are supported by extensive and well-designed experiments. This method has been validated across multiple architectures (Qwen, Pythia, Llama) and scales, demonstrating its universality. The comparison with relevant baselines is thorough and effectively highlights the superiority of the proposed technique.
- The paper is exceptionally well-written and presented. Figure 1 provides a clear and intuitive overview of the entire process. Figures 2, 3, and 4 are particularly effective in visualizing core results: the smooth performance interpolation of boomerang distillation contrasts with the poor or unstable performance of the baselines. These figures make the core claims convincing and easy to grasp.
- The ablation studies are also highly comprehensive, providing crucial support for the paper’s conclusions.

**Weaknesses:**

- The article mentions an inconsistency in performance between Llama-3.2-3B and the Qwen/Pythia Family. This limits the universality of the Boomerang Distillation method, suggesting that its successful application may require non-trivial, model-specific analysis—slightly undermining the "zero-shot" simplicity of the patching step.
- The article notes in Appendix E.1 that there is a lower bound for model compression when using this method. However, consider a scenario where we aim to distill DeepSeek-R1, a 671B model: in practice, we can obtain a 1.5B model through data distillation. This size reduction far exceeds the experimental lower bound of the method, revealing certain limitations of Boomerang Distillation.

**Questions:**

- Would the same effect still hold if simple modifications are made to the student model? For instance, the internal structure of the student model might differ from that of the teacher model—such as altered activation functions—even when their hidden dimensions are identical. If the current effect remains unchanged, what conclusions could be drawn from this?
- In Section 3.2 of the article, it is mentioned that low-quality datasets lead to the need for direct model distillation methods like Boomerang Distillation. Could openly available high-quality datasets be used to validate this conclusion?

---

> ### Author Response · Authors · 2025-11-21
> **Response to reviewer Cce4**
>
> We thank the reviewer for their feedback and positive comments about the novelty and efficiency of boomerang distillation, as well as how the paper is “exceptionally well-written and presented” and provides “extensive and well-designed experiments.” In the following, we address their remaining questions and concerns.
>
> **Response to Weaknesses:**
>
> > The article mentions an inconsistency in performance between Llama-3.2-3B and the Qwen/Pythia Family. This limits the universality of the Boomerang Distillation method, suggesting that its successful application may require non-trivial, model-specific analysis—slightly undermining the "zero-shot" simplicity of the patching step.
>
> We would like to provide a slight clarification to this point: regardless of the architecture, the patching step is still “zero-shot”. This is because the teacher to student layer mapping is predetermined by the initialization of the student model. However, the reviewer raises a valid question about universal applicability. We note that as discussed in Appendix I of the paper, the highlighted discrepancies are likely caused by a low cosine similarity between the first and last layers of the first teacher block in the Llama model, which could not be overcome with our limited training budget of 2B tokens. We hypothesize that if the small model is distilled for longer, for example, with Chinchilla optimal tokens (around 20x our compute budget), the alignment loss would be sufficient to enforce similarity in the Llama model, and our modifications to the initialization strategy would not be required.
>
> We agree (and state in Appendix I) that the Llama results suggest that a best practice in this limited compute budget distillation setting is to do some examination of the teacher to decide how the student is initialized. This step is very lightweight and simply consists of computing a forward pass for a very small calibration set (we used 128 documents) and creating a pairwise cosine similarity matrix comparing the activations at each layer of the teacher to ensure that the first and last layers in each teacher block have relatively high cosine similarity (Appendix I). We note that the pipeline of selecting teacher blocks based on pairwise cosine similarities between layers could be completely automated without requiring any manual ad hoc analysis (however, we implemented a manual strategy for each model for simplicity).
>
> We have updated our limitations section of the revised paper to include a discussion of this point.
>
> > The article notes in Appendix E.1 that there is a lower bound for model compression when using this method. However, consider a scenario where we aim to distill DeepSeek-R1, a 671B model: in practice, we can obtain a 1.5B model through data distillation. This size reduction far exceeds the experimental lower bound of the method, revealing certain limitations of Boomerang Distillation.
>
> We note that similarly to the response to Weakness 1 above, the lower bound in model compression is likely due to the low cosine similarity between the first and last layer of some teacher blocks when we initialize the student model with a naive every nth teacher layer setup. However, as stated in the response to Weakness 1, this could likely be overcome with longer distillation training.
>
> In our setting with limited training tokens, we *ran a new experiment to test the lower bound of the student model size.* Similarly to the Llama experiments, in order to further compress the student model, we examined the average pairwise layer cosine similarity of the activations of the model and initialized the student by manually selecting layers such that the first and last layer in each teacher block did not have vastly different activations. We provide the details of this selection method in Appendix E.2 of the revised paper.
>
> We ran additional experiments to determine how small the student model can be while preserving interpolation behavior and found that we can compress the student model 8.7x while still observing smooth interpolation (see Appendix E.2 of the revised paper). In the paper, we chose to use a model that keeps half of the teacher layers in order to maintain maximum granularity with respect to the size of the interpolated models. With these smaller student models, we can only generate M-1 intermediate models (where M is the number of student layers), since each student layer is swapped with its entire corresponding block of teacher layers in order to maintain alignment.

---

> > ### Author Response · Authors · 2025-11-21
> > **Response to reviewer Cce4 questions**
> >
> > **Response to Questions**
> >
> > > Would the same effect still hold if simple modifications are made to the student model? For instance, the internal structure of the student model might differ from that of the teacher model—such as altered activation functions—even when their hidden dimensions are identical. If the current effect remains unchanged, what conclusions could be drawn from this?
> >
> > This is an interesting question and is a limitation of our current experiments. We leave the implementation to future work, since it requires modifying our current initialization in order to extend boomerang distillation to heterogeneous internal structures. Specifically, it is unclear whether a student layer can be directly initialized by copying a teacher layer and modifying the activation function, since the different activation function may cause compounding errors or misalignment in the student model. This is an exciting avenue for future work, and we have included additional discussion on the limitations with respect to student and teacher internal structure and architecture in the limitations section of the paper.
> >
> > > In Section 3.2 of the article, it is mentioned that low-quality datasets lead to the need for direct model distillation methods like Boomerang Distillation. Could openly available high-quality datasets be used to validate this conclusion?
> >
> > To provide a slight clarification, we state that low-quality datasets can lead to catastrophic forgetting in distillation, rather than that our approach is needed to resolve the issue of low-quality datasets. We have revised Section 3.2 to make this more clear. We do not have access to the proprietary datasets used to train Qwen3 and Llama3.2 models. However, we note in Section 3.2 and Appendix G that for Pythia-2.8B (which has also been trained on The Pile), standard distilled models have slightly better performance than interpolated models created with boomerang distillation. This is likely because we perform distillation training using the same dataset used to train the teacher model, so we do not observe catastrophic forgetting in this setup.

---

> > > ### Comment · Reviewer_Cce4 · 2025-11-25
> > > **Official Comment by Reviewer Cce4**
> > >
> > > Thank you! You have addressed all my concerns and I will keep my positive rating.

---

> > > > ### Author Response · Authors · 2025-11-26
> > > >
> > > > Thank you for your time and consideration.

---

### Official Review · Reviewer_dyqi · 2025-10-30

**Soundness:** 3
**Presentation:** 3
**Contribution:** 2
**Rating:** 6
**Confidence:** 4

**Summary:**

This paper is concerned with knowledge distillation for dynamic inference. It observes that by treating the layer-dropped LM as the student LM and the original LM as the teacher LM, the student LM could dynamically adjust the inference budget by inserting dropped layers again into the student LM. With the inserted layers, the performance of the student LM could also be improved. The method proposed is termed boomerang distillation in this paper. The method enables a fine-grained model family with only one pass of distillation, reducing the expected cost.

**Strengths:**

1. It is actually a very counter-intuitive observation that the layer-dropped student could be recovered by adding the dropped layers back.
2. The one-pass distillation essentially enables efficient model family derivation, which may serve as a cost-effective way in both academia and industry.

**Weaknesses:**

1. The experiments are rather limited to model scales smaller than 10B, which might constrain the application of the method to larger-scale LMs.
2. In Figure 2, there seems to be a sudden performance improvement in generation accuracy along the insertion of layers, however, the improvement in classification accuracy is rather smooth. Further elaborations on the phenomenon should be attached here.

**Questions:**

N/A

---

> ### Author Response · Authors · 2025-11-21
> **Response to reviewer dyqi**
>
> We thank the reviewer for their comments and their positive feedback about how boomerang distillation “enables efficient model family derivation” and can be “cost-effective […] in both academia and industry.” In the following, we will address their concerns and questions.
>
> **Response to Weaknesses:**
>
> > The experiments are rather limited to model scales smaller than 10B, which might constrain the application of the method to larger-scale LMs.
>
> We have added an additional experiment replicating our results in Figure 2 with Pythia-12B and included it in Appendix Section F of the revised paper (Figure 14). We are currently running experiments on Qwen3-14B-Base and will update the paper draft and add an updated comment in the next few days when we have the results. We find that for the larger Pythia model, boomerang distillation also produces smooth interpolation behavior. With these new experiments, we have shown that boomerang distillation occurs across three orders of magnitude of model size (100M to 10B scale).
>
> > In Figure 2, there seems to be a sudden performance improvement in generation accuracy along the insertion of layers, however, the improvement in classification accuracy is rather smooth. Further elaborations on the phenomenon should be attached here.
>
> We agree that this is an interesting phenomenon and have revised Section 3.1 of the paper to provide additional discussion of the trend in generation versus classification performance. This discrepancy has been observed in prior work; in particular, the ShortGPT paper [1] mentions that layer pruning has a more negative effect on the performance of generative tasks than classification tasks, a finding we confirm in Figure 7. The authors of the ShortGPT paper hypothesize that this is because accumulated errors are a larger problem in generative tasks than in classification tasks. However, we note that boomerang distillation maintains much higher generation performance for smaller models as compared to pruning methods like ShortGPT. We believe this is because interpolating between the small and large model provides additional information as compared to simply compressing the large model.
>
> [1] Men et. al. ShortGPT: Layers in Large Language Models are More Redundant Than You Expect, 2024.

---

> > ### Author Response · Authors · 2025-11-26
> > **Qwen3-14B results**
> >
> > We have added the results for boomerang distillation using Qwen3-14B-Base in Figure 12 of the revised paper. We find that for Qwen3-14B-Base, boomerang distillation produces smooth interpolation behavior, demonstrating that boomerang distillation occurs for larger models with tens of billions of parameters. We note that after adding this figure, the Pythia-12B results referenced in the above comment are in Figure 15.
> >
> > We hope our responses have resolved your concerns and would greatly appreciate any other comments or suggestions that you may have.
> >
> > Thank you for your review and consideration.

---

### Official Review · Reviewer_s5vT · 2025-11-02

**Soundness:** 4
**Presentation:** 4
**Contribution:** 3
**Rating:** 6
**Confidence:** 5

**Summary:**

This work explores an interesting phenomenon termed Boomerang Distillation. The authors discover that, under certain constraints, adding back layers from a distilled model can produce a spectrum of intermediate model sizes without any additional training. They further show that these interpolated models achieve performance comparable to standard distilled models of the same size.

**Strengths:**

- The phenomenon presented is both interesting and novel.
- The presentation is clear, and the experimental results are comprehensive and well-executed.

**Weaknesses:**

- The paper lacks an in-depth explanation of the phenomenon. While a rigorous theoretical analysis may be unrealistic, including some qualitative observations or a proof-of-concept experiment to illustrate the underlying rationale would be beneficial.
- In the related work section, the authors compare their approach to model interpolation and claim that prior interpolation methods operate solely in weight space. However, interpolating across different architectures can also be regarded as a special case of weight-space interpolation (where certain weight matrices are zero). Moreover, LlamaFlex[1] discusses interpolation between models of different sizes in its “Router Interpolation” section, which seems relevant and worth acknowledging.
- This work focuses exclusively on constructing student models through layer dropping. It would be valuable to discuss whether the observed phenomenon also applies to width or head pruning. While interpolation in these cases may be more challenging, such discussion would provide a more complete perspective.

[1] LLaMaFlex: Many-in-one LLMs via Generalized Pruning and Weight Sharing

**Questions:**

None

---

> ### Author Response · Authors · 2025-11-21
> **Response to reviewer s5vT**
>
> We thank the reviewer for their feedback and appreciate their positive comments on how the phenomenon is “both interesting and novel” with “clear” presentation and “comprehensive” experiments.
>
> **Response to Weaknesses**
>
> > The paper lacks an in-depth explanation of the phenomenon. While a rigorous theoretical analysis may be unrealistic, including some qualitative observations or a proof-of-concept experiment to illustrate the underlying rationale would be beneficial.
>
> We agree that theoretical analysis of the phenomenon would be valuable and the question of exactly why boomerang distillation works is interesting. The main intuition behind our approach is that through our initialization and distillation procedure, we ensure that each student layer approximates the functionality of its corresponding teacher block (i.e., a single student layer is aligned to a block of teacher layers). Since each student layer approximates a teacher block, by replacing student layers with their corresponding teacher blocks we arrive at a *better approximation* of the teacher. We have revised Section 2 of the paper to include an explanation of this intuition. We have also added an **additional proof-of-concept experiment** (see Appendix K) showing that for multiple models, the average cosine similarity between the outputs of the last student and teacher layer increases as more student layers are patched. This supports our claim that patching enables the student to better approximate the teacher.
>
> > In the related work section, the authors compare their approach to model interpolation and claim that prior interpolation methods operate solely in weight space. However, interpolating across different architectures can also be regarded as a special case of weight-space interpolation (where certain weight matrices are zero). Moreover, LlamaFlex[1] discusses interpolation between models of different sizes in its “Router Interpolation” section, which seems relevant and worth acknowledging.
>
> We thank the reviewer for this reference. We have revised our related work section to more thoroughly discuss interpolation between models of different sizes, as well as LlamaFlex. LlamaFlex trains the teacher model in an elastic transformer architecture along with a Gumbel Softmax-based router and interpolates model sizes using the trained router. On the other hand, boomerang distillation trains a smaller student model using a standard knowledge distillation pipeline and interpolates model sizes by patching the distilled student with teacher layers, and does not require training a specialized router.
>
>
> > This work focuses exclusively on constructing student models through layer dropping. It would be valuable to discuss whether the observed phenomenon also applies to width or head pruning. While interpolation in these cases may be more challenging, such discussion would provide a more complete perspective.
>
> You are right that interpolating between student and teacher models with width or head pruning strategies is potentially more challenging than layer pruning. Below, we discuss more about the challenges with width and head pruning. First, if the teacher and student have different hidden dimensions due to width pruning, student patching cannot be applied out of the box because it would result in a mismatch in the output dimension of the hidden states. Next, in the head pruning case, it is unclear which heads to prune to limit performance degradation. Experimenting with boomerang distillation in these settings is an exciting avenue for future work, and we have included additional discussion on the limitations with respect to student and teacher architecture in the limitations section of the paper. We note that although the student layers’ architecture must match that of the teacher in our implementation, there is still a high degree of flexibility with respect to the size of the student, since we show in new experiments (Appendix E.2) that the student can be compressed up to 8.7x while still maintaining smooth interpolation behavior.

---

> > ### Author Response · Authors · 2025-11-26
> >
> > We hope our responses have resolved the weaknesses you raised and would appreciate any other comments or suggestions that you may have.
> >
> > Thank you for your review and consideration.

---

### Official Review · Reviewer_7Ngh · 2025-11-02

**Soundness:** 3
**Presentation:** 3
**Contribution:** 3
**Rating:** 4
**Confidence:** 4

**Summary:**

This paper introduces Boomerang Distillation, a novel approach for knowledge distillation that allows zero-shot model size interpolation. The key idea is to train the teacher and student jointly in a bidirectional distillation loop (“boomerang” scheme), where gradients from both directions are exchanged in a dynamically balanced manner. The method achieves smooth performance interpolation across different model scales and architectures (e.g., ViT and ResNet families), showing competitive results on ImageNet, CIFAR, and transfer benchmarks, while maintaining efficiency and data-free adaptability.
The paper introduces a technique termed boomerang distillation, in which a large “teacher” model is first distilled into a smaller “student” model, and then a family of intermediate-sized models is generated without further training by gradually re-inserting blocks from the teacher into the student. The resulting models interpolate in size (number of parameters) and in performance smoothly between student and teacher, and in many cases match or outperform traditional distilled models of the same size. The authors analyse the necessary conditions and compare the method to layer-pruning baselines, showing favorable results.

**Strengths:**

1.Enables flexible interpolation between different model sizes without retraining, which is practical for model deployment across devices.
2.Demonstrated across multiple architectures and datasets with consistent improvements and smooth interpolation curves.
3.The paper goes into details about what components are necessary: e.g., the alignment loss, the initialization from teacher weights, etc. They also compare against layer-pruning baselines. This helps the community understand when/why the method works.

**Weaknesses:**

1.It remains unclear how the method scales to much larger teacher models (tens or hundreds of billions of parameters) and whether the interpolation remains as clean in those regimes.
2.The paper focuses on language models. It would strengthen the work to show applicability in other modalities (vision, multimodal) or architectures for broader impact.
3.While no further training is required for each intermediate size, the approach still requires the initial distillation and alignment of a student model—which may itself be costly.
4.The method assumes that student and teacher share compatible hidden dimensions and layer structure to allow patching. This may limit generality to heterogeneous architectures or differently sized hidden layers.

**Questions:**

1.How sensitive is the method to the relative capacity of teacher and student (e.g., extreme size gaps)?
2.If the teacher and student have different hidden dimensions, can patching still be applied?
3.If the model is multimodal with fusion layers, could inserting teacher blocks disrupt cross-modal interactions, and how might you control or evaluate that risk?
4.What is the practical GPU cost increase compared to standard KD?

---

> ### Author Response · Authors · 2025-11-21
> **Response to reviewer 7Ngh**
>
> We thank the reviewer for their comments and positive feedback on how our approach is “novel”, “flexible”, and “practical.”
>
> **Summary clarification**
> We would like to clarify a few misunderstandings in the summary. First, the reviewer wrote that “the key idea is to train the teacher and student jointly in a bidirectional distillation loop (“boomerang” scheme), where gradients from both directions are exchanged in a dynamically balanced manner”. To clarify, in our approach we freeze the teacher model and only perform gradient updates on the student using alignment information from the teacher. In this setup, we do forward and backward passes on the student, but only perform forward passes on the teacher. As a result, the computational overhead for our distillation step is the same as standard distillation. A second misconception is regarding the datasets and architectures: the reviewer mentions interpolation “across different model scales and architectures (e.g., ViT and ResNet families), showing competitive results on ImageNet, CIFAR, and transfer benchmarks”. However, we focus our experiments on LLMs and do not include any vision datasets or models.
>
> **Response to Weaknesses**
>
> > 1. It remains unclear how the method scales to much larger teacher models (tens or hundreds of billions of parameters) and whether the interpolation remains as clean in those regimes.
>
> We have added an additional experiment replicating our results in Figure 2 with Pythia-12B and included it in Appendix Section F of the revised paper (Figure 14). We are currently running experiments on Qwen3-14B-Base and will update the paper draft and add an updated comment in the next few days when we have the results. We find that for the larger Pythia model, boomerang distillation also produces smooth interpolation behavior. With these new experiments, we have shown that boomerang distillation occurs across three orders of magnitude of model size (100M to 10B scale).
>
> > 2. The paper focuses on language models. It would strengthen the work to show applicability in other modalities (vision, multimodal) or architectures for broader impact.
>
> We thank the reviewer for this suggestion. We agree that applying boomerang distillation to other modalities would be a valuable contribution, but it is beyond the scope of this paper. We have highlighted extensions to other modalities as possible future directions in the limitations section of our updated manuscript.
>
> > 3. While no further training is required for each intermediate size, the approach still requires the initial distillation and alignment of a student model—which may itself be costly.
>
> The reviewer is correct in that boomerang distillation requires one initial knowledge distillation run, which can be computationally expensive. However, this computational cost has to be paid in standard knowledge distillation as well, and in contrast to standard KD, we get numerous intermediate models between the distilled student and teacher models without any additional training, enabling us to save enormous amounts of training compute. Our approach can thus be thought of as an instance of training cost amortization, since after the initial distillation training, all additional interpolated models can be created in a zero-shot manner. We ran new experiments to quantify the amount of compute saved with boomerang distillation as compared to individual distillation runs. We have clarified this in Appendix L and show in the table below (Table 3 in the revised paper) that boomerang distillation requires 14.53-19.17x less FLOPs than individually distilling intermediate models.
>
> | **Teacher Model** | **Models** | **FLOPS** | **Theoretical Compute Speedup** |
> |-------------|-------------|----------:|---------------------------------:|
> | Qwen3-4B-Base | Standard distillation | 8.27e20 | - |
> | | Boomerang distillation | 4.31e19 | 19.17x |
> | Pythia-2.8B | Standard distillation | 4.77e20 | - |
> | | Boomerang distillation | 2.80e19 | 17.01x |
> | Llama-3.2-3B | Standard distillation | 5.07e20 | - |
> | | Boomerang distillation | 3.49e19 | 14.53x |
>
> > 4. The method assumes that student and teacher share compatible hidden dimensions and layer structure to allow patching.
>
> We agree that this is a limitation of our approach and leave the implementation to future work, since extending boomerang distillation to heterogeneous architectures is challenging and requires significant modifications to the distillation and patching procedure. We have included additional discussion on the limitations with respect to student and teacher architecture in the limitations section of the paper. We note that although the student layers’ architecture must match that of the teacher, there is still a high degree of flexibility with respect to the size of the student, since we show in Appendix E.2 of the revised paper (see response to Question 1) that the student can be compressed up to 8.7x while still maintaining smooth interpolation behavior.

---

> > ### Author Response · Authors · 2025-11-21
> > **Response to Reviewer 7Ngh questions**
> >
> > **Response to Questions**
> >
> > > 1. How sensitive is the method to the relative capacity of teacher and student (e.g., extreme size gaps)?
> >
> > We ran additional experiments to determine how small the student model can be while preserving interpolation behavior and found that **we can compress the student model 8.7x while still observing performance interpolation** (see Appendix E.2 of the revised paper). We note that in the paper, we chose to use a model that keeps half of the teacher layers in order to maintain maximum granularity with respect to the size of the interpolated models. With these smaller student models, we can only generate M-1 intermediate models (where M is the number of student layers), since each student layer is swapped with its entire corresponding block of teacher layers in order to maintain alignment.
> >
> > > 2. If the teacher and student have different hidden dimensions, can patching still be applied?
> >
> > If the teacher and student have different hidden dimensions, patching cannot be applied out-of-the-box without significant modifications, since layers from models with different hidden dimensions cannot be naively concatenated without some form of mapping from one hidden dimension to the other. As noted in the response to Weakness 4, we agree that this is a limitation of our approach and added additional discussion on this in the limitations section of the revised paper.
> >
> > > 3. If the model is multimodal with fusion layers, could inserting teacher blocks disrupt cross-modal interactions, and how might you control or evaluate that risk?
> >
> > Thank you for the question. Our approach relies on aligning the representations of the student and teacher models, so it should be model-agnostic in principle. Could you clarify what you mean by "fusion layers" in the context of multi-modal models and what the additional risk would be in this setting?
> >
> > > 4. What is the practical GPU cost increase compared to standard KD?
> >
> > As mentioned in Weakness 3 above, boomerang distillation requires a single distillation run to construct multiple models. As a result, distilling the student model requires exactly the same amount of compute as distilling an equivalent sized model in a standard distillation setup, but produces up to N/2 total intermediate models without any additional training. Here, N is the number of teacher layers.

---

> > > ### Author Response · Authors · 2025-11-26
> > > **Qwen3-14B results**
> > >
> > > We have added the results for boomerang distillation using Qwen3-14B-Base in Figure 12 of the revised draft. We find that for Qwen3-14B-Base, boomerang distillation produces smooth interpolation behavior, demonstrating that boomerang distillation occurs for larger models with tens of billions of parameters. We note that after adding this figure, the Pythia-12B results referenced in the above comment are in Figure 15.
> > >
> > > We hope our responses have addressed your concerns and welcome any other comments or suggestions that you may have.
> > >
> > > Thank you for your review and consideration.

---

### Author Response · Authors · 2025-11-21
**General comment**

We thank the reviewers for their positive and helpful feedback. We appreciate how reviewers thought boomerang distillation was a "novel” [7Ngh, s5vT, Cce4] and "counter-intuitive” [dyqi] phenomenon that is “highly practical” [Cce4]. They found our paper to be “exceptionally well-written” [Cce4] and “clear” [s5vT], and our experiments “highly comprehensive” [Cce4] and “well-executed” [s5vT]. Below, we address the main concerns brought up by the reviewers.

1. **Scaling to larger teacher models**

Reviewers 7Ngh and dyqi noted that our experiments were limited to models below 10B parameters. We ran new experiments on Pythia-12B and are running experiments on Qwen3-14B-Base, which will be added to the paper in the next few days. The results on Pythia-12B,  shown in Figure 14 of the revised paper, demonstrate that the boomerang distillation phenomenon is consistent even for larger model sizes. With these new experiments, we have now shown that boomerang distillation occurs across three orders of magnitude of model size (100M to 10B scale).

2. **Lower bound on compression rate of the teacher**

Reviewers 7Ngh and Cce4 asked how much the teacher can be compressed. We added new experiments in Appendix E.2 showing that we can compress the teacher model up to 8.7x to initialize the student by manually choosing which layers to drop. We note that in the paper, we largely focus on the setting where the student model is initialized with every other teacher layer rather than maximally compressing the student, since initializing with every other layer provides maximum granularity in the possible interpolated models.

3. **Architectural constraints of the student model**

Reviewers 7Ngh, s5vT, and Cce4 correctly note that the student’s architecture is constrained to have the same types of layers and same hidden dimension as the teacher model. While we agree this is a limitation, boomerang distillation offers significant computational advantages and a high degree of flexibility: we can create interpolated models between the student and the teacher of different sizes without any additional training. We compare boomerang distillation to ShortGPT and LaCO, two well-known pruning methods that also require the student to have the same architecture as the teacher. Our results show that we significantly outperform them, especially at high pruning rates, while offering the same degree of flexibility. Extending boomerang distillation to heterogeneous student-teacher pairs with different widths and head dimensions is an exciting future direction. We have updated our limitations section to highlight this potential restriction and future work.

4. **Additional clarifications**

In response to reviewer s5vT’s request for an in-depth explanation and proof-of-concept experiment demonstrating why boomerang distillation works, we have provided **additional proof-of-concept results** in Appendix K showing that as the student layers are patched, the student model better approximates the last layer activations of the teacher. We have also revised our discussion in Section 2 to include an explanation of the intuition behind boomerang distillation.

Additionally, we have provided new experimental results in Appendix L showing the computational resources required for boomerang distillation (14.53-19.17x less than distilling individual models) as requested by Reviewer 7Ngh.

We thank the reviewers for their other comments requesting clarifications on boomerang distillation with other modalities [7Ngh], related work and router interpolation [s5vT], generation versus classification accuracy trends [dyqi], setup for Llama versus Qwen/Pythia [Cce4], and experimental datasets [Cce4]. We have revised the writing in the limitations, related work, and experimental sections to address these concerns.

Again, we are grateful to the reviewers for their constructive feedback, which led to additional experiments and revisions that have made the paper stronger and clearer. Between this general response and the individual responses below, we believe we have addressed all key concerns, and we welcome any further discussion or clarification.

---

> ### Author Response · Authors · 2025-11-26
> **Additional Qwen3-14B-Base results**
>
> We have added the results for boomerang distillation using Qwen3-14B-Base in Figure 12 of the revised draft (we note that after adding this figure, the Pythia-12B results referenced in the above comment are in Figure 15). We find that for Qwen3-14B-Base, boomerang distillation produces smooth interpolation behavior, demonstrating that boomerang distillation occurs for larger models with tens of billions of parameters.

---

### Meta-Review · Area_Chair_T8au · 2025-12-22

**Summary:**

Reviewer 7Ngh expressed concerns on the compatibility on heterogeneous architectures or differently sized hidden size. Moreover, Reviewer 7Ngh provided inaccurate summary about this paper. Reviewer s5vT pointed out this paper doesn't analyze this phenomenon in depth. Reviewer dyqi argued that this paper should supplement experiments on models larger than 10B. Reviewer Cce4 pointed out inconsistent performance between Llama-3.2-3B and the Qwen, thus limiting the universality of the Boomerang Distillation.

Furthermore, AC find that there are also several previous works [R1, R2] that are highly related to Boomerang Distillation, which also allows block-wise interpolation.

[R1] Bai et al. Few Shot Network Compression via Cross Distillation. AAAI 2020.

[R2] Shen et al. Progressive Network Grafting for Few-Shot Knowledge Distillation. AAAI 2021.

**Reviewer Concerns:**

The direct application on heterogeneous architectures is a limitation of this paper. AC believes that this issue doesn't affect the main contribution of this paper. Authors partially explained the smooth performance scaling and supplement the experiments 10B+ model. The issue about inconsistent performance is also partially resolved.

**Reviewer Scores:**

Reviewer 7Ngh: 4;

Reviewer s5vT: 6;

Reviewer dyqi: 6;

Reviewer Cce4: 8

---

### Decision · Program_Chairs · 2026-01-26

Accept (Poster)